# Is two-point method a valid and reliable method to predict 1RM? A systematic review

**Zongwei Chen, Zheng Gong, Liwen Pan, Xiuli Zhang**\*

School of Physical Education and Sports Science, South China Normal University, Guangzhou, Guangdong Province, China

\* hzhzhz411@126.com

## Abstract

This systematic review aimed to evaluate the reliability and validity of the two-point method in predicting 1RM compared to the direct method, as well as analyze the factors influencing its accuracy. A comprehensive search of PubMed, Web of Science, Scopus, and SPORT-Discus databases was conducted. Out of the 88 initially identified studies, 16 were selected for full review, and their outcome measures were analyzed. The findings of this review indicated that the two-point method slightly overestimated 1RM (effect size = 0.203 [95%CI: 0.132, 0.275]; $P < 0.001$); It showed that test-retest reliability was excellent as long as the test loads were chosen reasonably (Large difference between two test loads). However, the reliability of the two-point method needs to be further verified because only three studies have tested its reliability. Factors such as exercise selection, velocity measurement device, and selection of test loads were found to influence the accuracy of predicting 1RM using the two-point method. Additionally, the choice of velocity variable, 1RM determination method, velocity feedback, and state of fatigue were identified as potential influence factors. These results provide valuable insights for practitioners in resistance training and offer directions for future research on the two-point method.

**Data Availability Statement:** All relevant data are within the paper and its Supporting Information files.

**Funding:** the Multifunctional Integrated Digital Strength and Conditioning Training Laboratory

## Introduction

Resistance training (RT) serves as a pivotal modality for athletes to optimize their athletic prowess and safeguard against injuries [1, 2]. Moreover, it represents a potent avenue for individuals in the general population to ameliorate their overall health status [3]. To elicit desirable physiological adaptations through RT, meticulous control over various training variables, encompassing exercise selection and sequencing, intensity and volume, rest intervals, and training frequency, assumes paramount importance. Significantly, among these variables, training intensity assumes a preeminent role in augmenting strength levels [4–6].

The assessment of %1RM (i.e., the percentage of the maximum load that can be lifted once with full range of motion in a given exercise) has conventionally served as the "gold standard" for determining strength training intensity [7]. However, direct determination of 1RM is fraught with challenges, including time constraints, susceptibility to injuries, and lack of real-time assessment [8–10]. In order to overcome these limitations, numerous indirect methods

jointly constructed by South China Normal University and Beijing Yanding Huachuang Sports Development Co., Ltd (Grant number GDSJYT2022035)——receiver Xiuli Zhang. The funders had no role in study design, data collection and analysis, decision to publish, or preparation of the manuscript.

**Competing interests:** The authors have declared that no competing interests exist.

have been proposed to estimate 1RM, among which the load-velocity relationship (LVR) has gained considerable attention [11–13]. In a seminal study conducted by González-Badillo et al. in 2010 [13], it was demonstrated that an exceptionally strong inverse relationship ($R^2 = 0.98$) exists between movement velocity and training intensity (%1RM) during maximal effort exertion in the concentric phase, a correlation that remains stable even after a period of training. As a result, LVR has gradually emerged as a valuable tool in RT to quantify %1RM and 1RM over the past decade [14–19]. Researchers contend that LVR offers a time-efficient, safe, and real-time alternative to direct methods for assessing 1RM.

The regression models utilized for LVR analysis include linear and polynomial regression. In earlier studies, researchers commonly employed polynomial regression to establish LVR, typically requiring the assessment of movement velocity across 5–9 different loads [13, 20, 21]. In recent years, investigations have indicated no significant disparity in goodness of fit ($R^2$) between linear and polynomial regression models for LVR [22–24]. Consequently, both linear and polynomial models are considered viable approaches for defining LVR. Hence, based on the mathematical principle of "two points and one line", a definition of LVR utilizing only two loads has been proposed [25, 26]. This two-point method enables rapid prediction of maximum dynamic strength by measuring movement velocity at two loads and knowing the velocity of 1RM ($V_{1RM}$) [25]. $V_{1RM}$ can be determined directly or extracted from previous studies. It should be acknowledged that the two-point method allows for quick 1RM prediction compared to the direct method. However, its ability to accurately reflect the maximum dynamic strength level remains uncertain. Some studies have reported a strong correlation between 1RM values predicted by the two-point method and those measured by the direct method [19, 26]. Conversely, there have been reports indicating that 1RM values predicted by the two-point method were significantly higher than those measured by the direct method [27]. What accounts for these contradictory findings? To the best of our knowledge, the accuracy of LVR varies depending on factors such as the velocity measurement device used, velocity variable, exercise type, and execution technique [28–31]. Given the lack of a systematic review analyzing the impact of these factors on the accuracy of 1RM predicted by the two-point method, we were intrigued by the reasons underlying the aforementioned conflicting results.

Therefore, the main purpose of this systematic review was to analyze the reliability and validity of the two-point method in 1RM prediction compared with the direct method. Additionally, we aimed to examine the factors that could potentially influence the accuracy of the two-point method in 1RM prediction.

## Materials and methods

### Literature search strategy

The literature search was conducted on electronic databases until June 10, 2023. Two reviewers (Chen and Gong) independently searched the following databases: PubMed, Web of Science, SPORTDiscus, and Scopus. The search strategy used the following terms, which were adapted for each database and applied to the title and abstract search: ("two-point method*" OR "two-point" OR "2-point method*" OR "2-point" OR "two-load method*" OR "two loads" OR "2-load method*" OR "2 loads") AND ("one-repetition maximum" OR "1RM" OR "maximal dynamic strength" OR "1-RM"). The search results were collected and imported into a reference manager (EndNote X9, Thomson Reuters, Philadelphia, PA, USA).

### Screening process

Two independent reviewers (Chen and Gong) conducted the screening of studies retrieved from each database. The screening process followed the following approach: 1) initial selection

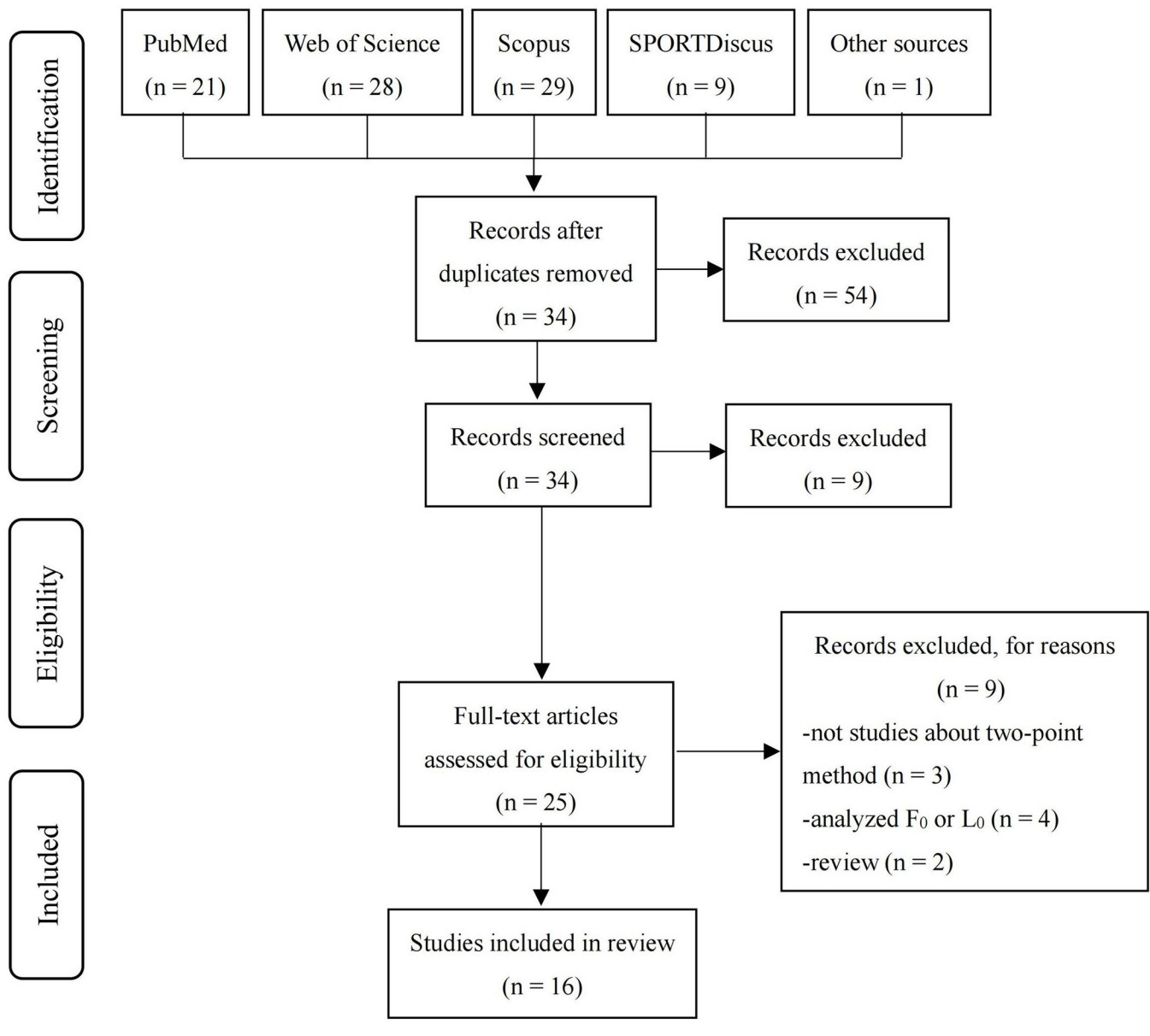

**Fig 1. Search and screening procedure.**

based on title and abstract, with removal of duplicates; 2) comprehensive examination of the remaining studies, excluding those that were determined to be outside the scope of the present review (Fig 1). Any discrepancies between the reviewers were resolved through discussion. If consensus could not be reached, a third reviewer was consulted.

## Eligibility criteria

Inclusion criteria for the review were as follows: 1) studies published in English; 2) journal articles with full-text availability; 3) studies involving individuals of any age, gender, and RT experience; 4) studies including individuals without musculoskeletal injuries; 5) studies that analyzed the validity and/or reliability of 1RM predicted by the two-point method.

All forms of grey literature (e.g., conference papers, theses, reports, abstracts, proceedings, etc.) were be excluded.

Consistent with previous research, 1RM was defined as the maximum load that can be lifted once with full range of motion in a given exercise [7], and the two-point method was defined as a method for predicting 1RM that utilizes the velocity values of two distinct loads and the velocity value of 1RM in LVR [32].

## Data extraction

Two independent reviewers (Chen and Gong) utilized standardized forms in Microsoft Excel 2019 (Microsoft Corporation, USA) to collect data from all included studies. The collected data encompassed various aspects, including author, sample characteristics (age, gender, height, body mass and RT experience), specific exercise details, execution technique, velocity measurement device, velocity variable, 1RM values obtained through the direct method and the two-point method, test load employed in the two-point method. Validity indicators included $P$-value, Pearson correlation coefficient (r), effect size (ES, Cohen's d or Hedge's g), systematic bias and random error (SB±RE) and heteroscedasticity r-square ($R^2$). Reliability indicators included interclass correlation coefficient (ICC) and within-subjects coefficient of variation (CV). We did not consider including $L_0$ (load at zero velocity) and $F_0$ (force at zero velocity) because the lack of clear physiological significance in $L_0$ and the inability of $F_0$ to represent the lift performed by subjects at full range of motion [33, 34].

We used non-negative ES to reflect the degree of difference between the predicted 1RM and the actual 1RM to avoid bias in the results, and the ES from different studies were synthesized using Comprehensive Meta-Analysis V3 software (Biostat, New Jersey, USA) [35]. The results were presented in the form of a forest plot using Microsoft Excel 2019, thereby providing a visual representation of the validity and influencing factors associated with the two-point method. The criteria for interpreting the magnitude of ES were as follows: trivial ($< 0.20$), small (0.20–0.59), moderate (0.60–1.19), large (1.20–2.00), and very large ($> 2.00$) [36]. It should be noted that an ES closer to zero indicates higher validity. An acceptable ES was considered to be less than 0.2. The criteria for interpreting the magnitude of ICC were as follows: poor ($< 0.50$), moderate (0.50–0.75), good (0.75–0.90), and excellent ($> 0.90$) [37]. An acceptable ICC was considered to be higher than 0.75. The criteria for interpreting r were as follows: trivial ($< 0.09$), small (0.10–0.29), moderate (0.30–0.49), large (0.50–0.69), very large (0.70–0.89), nearly perfect (0.90–0.99), and perfect (1.00) [36].

Any discrepancies between the reviewers regarding the selection of indicators were resolved through discussion. If consensus could not be reached, the input of a third reviewer was sought.

## Dealing with missing data

We employed the following measures to address missing data: 1) statistical methods were utilized to calculate missing values based on the reported data. For data where actual 1RM and absolute difference was provided, we calculated predicted 1RM using "$M_{1+2} = M_1 + M_2, SD_{1+2} = \sqrt{SD_1^2 + SD_2^2}$". For data where 1RM was provided for both males and females, but not for the overall 1RM, we used "$M_1 \& 2 = \frac{\sum n_i \bar{X}_i}{\sum n_i}, SD_1 \& 2 = \sqrt{\frac{\sum n_i s_i^2 + \sum n_i d_i^2}{\sum n_i}}$", to calculate missing data; 2) missing data were extracted from figures using Originpro 2021 software (OriginLab, USA); 3) Corresponding authors were contacted to obtain any missing data.

It should be noted that we solely performed statistical calculations to address missing data and refrained from making adjustments to the associated data, as such adjustments could potentially introduce additional errors.

**Table 1. Downs and black modified checklist quality assessment.**

| Author | Q1 | Q2 | Q3 | Q6 | Q7 | Q10 | Q11 | Q16 | Q18 | Q20 | Total (/10) | Quality |
|---|---|---|---|---|---|---|---|---|---|---|---|---|
| García-Ramos et al. [26] | 1 | 1 | 1 | 1 | 0 | 0 | 0 | 1 | 1 | 1 | 7 | Moderate |
| García-Ramos et al. [19] | 1 | 1 | 1 | 1 | 0 | 1 | 0 | 1 | 1 | 1 | 8 | Moderate |
| Pérez-Castilla et al. [42] | 1 | 1 | 0 | 1 | 1 | 1 | 0 | 1 | 0 | 1 | 7 | Moderate |
| Caven et al. [43] | 1 | 1 | 0 | 1 | 0 | 0 | 1 | 1 | 0 | 1 | 6 | Moderate |
| Pérez-Castilla et al. [44] | 1 | 1 | 0 | 1 | 0 | 0 | 0 | 1 | 1 | 1 | 6 | Moderate |
| Fernandes et al. [45] | 1 | 1 | 0 | 1 | 0 | 1 | 0 | 1 | 0 | 1 | 6 | Moderate |
| Janicijevic et al. [46] | 1 | 1 | 1 | 1 | 0 | 0 | 0 | 1 | 0 | 1 | 6 | Moderate |
| Perez et al. [47] | 1 | 1 | 0 | 1 | 0 | 1 | 0 | 1 | 1 | 1 | 7 | Moderate |
| Pérez-Castilla et al. [48] | 1 | 1 | 0 | 1 | 0 | 0 | 0 | 1 | 1 | 1 | 6 | Moderate |
| Aidar et al. [49] | 1 | 1 | 0 | 1 | 0 | 0 | 0 | 1 | 1 | 1 | 6 | Moderate |
| Çetin et al. [50] | 1 | 1 | 0 | 1 | 0 | 1 | 0 | 1 | 1 | 1 | 7 | Moderate |
| Jiménez-Alonso et al. [51] | 1 | 1 | 1 | 1 | 1 | 0 | 0 | 1 | 1 | 1 | 8 | Moderate |
| Jukic et al. [52] | 1 | 1 | 1 | 1 | 0 | 1 | 0 | 1 | 1 | 1 | 8 | Moderate |
| Macarilla et al. [27] | 1 | 1 | 1 | 1 | 0 | 0 | 0 | 1 | 1 | 1 | 7 | Moderate |
| Soriano et al. [53] | 1 | 1 | 1 | 1 | 0 | 1 | 0 | 1 | 1 | 1 | 8 | Moderate |
| Kjær et al. [54] | 1 | 1 | 1 | 1 | 0 | 1 | 1 | 1 | 1 | 1 | 9 | High |

## Quality assessment of included studies

Two reviewers (Chen and Gong) independently assessed the quality of the included studies using a modified version of the Downs and Black checklist [38] (Table 1). Discrepancies between the reviewers were resolved through discussion. In the event that a consensus could not be reached, a third reviewer was consulted.

In accordance with Fox et al. [39], we employed 10 out of the 27 criteria that were logically applicable to all types of studies included in this review, resulting in a maximum total score of 10. Previous studies indicated that essential descriptors for scoring "1" on question 3 included gender, age, height, body mass, RT experience and actual 1RM level of the subjects. Likewise, the Bland-Altman method was considered an essential descriptor for scoring "1" on question 18 [40].

As there were no reference ranges available for the modified checklist, we compared it to the Physiotherapy Evidence Database (PEDro) scale [41] (Same highest total score) for classification: high quality (9–10), moderate quality (5–8), and low quality (0–4).

## Results

### Study selection

The initial search retrieved a total of 87 studies from the electronic database search, with an additional study identified through other sources (reference lists) (Fig 1). After removing duplicates, 34 titles and abstracts were screened, resulting in 25 potentially eligible studies. Following the full-text screening, a total of 16 studies were deemed suitable for inclusion in this systematic review.

### Quality of included studies

The quality score of the included studies assessed by Downs and Black modified checklist was (Mean ± Standard deviation [M±SD] = 7.00±0.97, range [6–9]) (Table 1). No study was excluded due to "low quality".

## Study characteristics

A total of 389 participants (335 [86.1%] males and 54 [13.9%] females) were included in this systematic review (Table 2). Among the studies, 15 investigated young-only samples [19, 26, 27, 42–44, 46–54] while one study included both young and middle-aged participants [45]. Four studies did not specify whether the subjects had RT experience [42, 44, 47, 48], while the remaining studies reported that the subjects had varying years of RT experience [19, 26, 27, 43, 45, 46, 49–54]. The participants consisted of three categories: RT enthusiasts (n = 149 [38.3%]), collegiate sports science students (n = 144 [37.0%]) and athletes (n = 96 [24.7%]).

A total of 8 exercises were investigated, including upper-limb exercises such as bench press [26, 27, 42–47, 49, 51], prone bench pull [19], bent-over-row [45], seated cable row [48], lat pulldown [48] and overhead press [53], as well as lower-limb exercises such as squat [27, 43, 50, 54] and deadlift [50, 52]. These exercises were performed using three types of modalities: free weight, Smith machine and machine. Variations in these exercises were also observed, such as bench press with different grip widths [44] and deadlift performed with or without lifting straps [52]. Two execution techniques were employed: eccentric-concentric (EC) and concentric-only (CO). In addition, tests performed under fatigue or non-fatigue conditions were also analyzed [54].

Five different types of velocity measurement devices were utilized in the included studies: linear position transducers (LPT) [19, 26, 27, 42, 43, 44, 46–52, 54], camera-based optoelectronic device (CBOD) [42], inertial measurement units (IMU) [42], smartphone application (APP) [42, 48] and rotary encoder [45]. All 16 studies used mean velocity (MV, mean velocity value from the start of the concentric phase until the load reaches the maximum height [29]) as the velocity variable to predict 1RM [26, 27, 19, 42–54], one study used mean propulsive velocity (MPV, mean propulsive velocity, mean velocity value from the start of the concentric phase until the acceleration of the load is lower than gravity [-9.8m/s$^2$] [26, 55]), and one study used peak velocity (PV, maximum instantaneous velocity value reached from the start of the concentric phase until the load reaches the maximum height) [54]. A total of 21 test load combinations were used to develop the two-point method model (Table 3).

Furthermore, four methods of 1RM determination based on two-point method were employed in the included studies. The first method involved testing the velocity values corresponding to two different loads and directly assessing the velocity value corresponding to 1RM ($V_{1RM}$) [19, 26, 27, 43, 46–48, 50, 52, 54], followed by calculating 1RM through linear regression. The second method entailed testing the velocity values corresponding to the two different loads and utilizing the $V_{1RM}$ value from previous studies to calculate 1RM [42, 44–46, 49, 51, 53]. The third method consisted of considering the velocity of the last repetition ($V_{last}$) in a specific load set as $V_{1RM}$ [46]. The fourth method involved calculating the value of 1RM by determining the intersection of the force-velocity relationship and the weight-velocity relationship [49].

## Study main outcomes

The primary outcomes regarding the validity and reliability of the two-point method in 1RM prediction were summarized in Table 3.

Upon conducting statistical calculations, it was determined that the two-point method yielded an overall ES of 0.203 (95%CI: 0.132, 0.275; $P < 0.001$), indicating a statistically significant but small difference when compared to the 1RM values obtained through the direct method.

Additionally, the overall ICC of the two-point method for predicting 1RM was found to be 0.797, denoting a good level of reliability albeit not excellent. Further elaboration on these findings could be found in the discussion section.

**Table 2. Study characteristics of included studies.**

| Authors | n, gender | Age (M±SD) Height (M±SD) Body mass (M±SD) | RT experience (M±SD) | Exercise (modality) | Execution Technique | Direct 1RM (M±SD) |
|---|---|---|---|---|---|---|
| García-Ramos et al. [26] | n = 30 (M), RT experience | 21.2±3.8 years 1.78 ± 0.07 m 72.3 ± 7.3 kg | ≥2 years | Bench press (SM) | CO EC | 76.3±13.0 kg (CO, T1) 76.0±12.7 kg (CO, T2) 80.5±13.7 kg (EC, T1) 80.9 ±14.2 kg (EC, T2) |
| García-Ramos et al. [19] | n = 26 (M), rowers and weightlifters | 20.5±2.9 years 1.76±0.07 m 75.7±9.3 kg | 6.1±3.9 years | Prone bench pull (FW) | CO | 89.8±13.4 kg (T1) 90.1±12.1 kg (T2) |
| Pérez-Castilla et al. [42] | n = 11 (M), RT experience | 22.5±1.9 years 1.75±0.06 m 75.2±7.2 kg | NA | Bench press (SM) | CO | 83.8±12.3 kg |
| Caven et al. [43] | n = 17 (F), netball players | 17.8 ± 1.3 years NA 69.1 ± 9.6 kg | ≥1 year | Bench press (FW)Squat (FW) | EC | 38.6±7.5 kg (BP) 86.5±14.7 kg (SQ) |
| Pérez-Castilla et al. [44] | n = 20 (M), CSSS | 22.5±3.7 years 1.78±0.06 m 77.9±13.1 kg | NA | Bench press (SM; CL, ME, WI and SE) | CO | 81.0±3.0 kg (Overall) 80.0±3.0 kg (CL) 83.0±3.0 kg (ME) 79.0±3.0 kg (WI) 81.0±3.0 kg (SE) |
| Fernandes et al. [45] | n = 40 (M; 20 young, 20 middle-aged), RT experience | 21.0±1.6 years (young) 42.6 ±6.7 years (middle-aged) NA NA 85.9±12.8 kg (young) 82.3 ±11.2 kg (middle-aged) | 4.5±1.1 years (young) 16.9±11.4 years (middle-aged) | Bench press (SM) Bent-over-row (SM) | EC (BP) CO (BOR) | NA |
| Janicijevic et al. [46] | n = 86 (M), CSSS | 20.9±4.2 years 1.73±0.05 m 74.3±15.6 kg | 1.3±2.4 years | Bench press (SM) | CO EC | 61.6±17.5 kg (CO) 66.3±18.3 kg (EC) |
| Perez et al. [47] | n = 20 (10M, 10F), RT experience | 28±8 years (M) 26±6 years (F) 177.22±9.92 cm (M) 162.61 ±5.03 cm (F) 97.32±20.09 kg (M) 63.35±4.58 kg (F) | NA | Bench press (FW) | EC | 90.23±45.18 (Overall) 129.55±30.47 (M) 50.91±7.89 (F) |
| Pérez-Castilla et al. [48] | n = 23 (12M, 11F), CSSS | 20.8±2.5 years (M) 20.2±1.1 years (F) 179.6±6.1 cm (M) 172.2±4.9 cm (F) 78.9±10.7 kg (M) 65.3±4.4 kg (F) | NA | Lat pulldown (Ma) Seated cable row (Ma) | CO | 62.8±19.6 kg (LPD, Overall) 59.9 ±18.8 kg (SCR, Overall) 78.1±14.0 kg (LPD, M) 74.4±14.2 kg (SCR, M) 46.1±7.3 kg (LPD, F) 44.1±6.2 kg (SCR, F) |
| Aidar et al. [49] | n = 15 (M), Paralympic Powerlifting athletes | 27.7±5.7 years NA 74.0±19.5 kg | 2.1±0.9 years | Bench press (FW) | CO | 113.0±31.3 kg |
| Çetin et al. [50] | n = 13 (M), RT experience | 23.6±4.2 years 179.6±7.1 m 80.2±8.9 kg | ≥2 years | Deadlift (FW) Squat (FW) | CO | 171.56±33.18 kg (deadlift) 148.59 ±36.03 kg (squat) |
| Jiménez-Alonso et al. [51] | n = 15 (M), CSSS | 20.5±3.0 years 1.75±0.06 m 74.3±8.8 kg | 1.6±0.9 years | Bench press (FW) | EC | 79.1±18.3 kg (VF) 79.4±17.3 kg (NVF) |
| Jukic et al. [52] | n = 18 (M), RT experience | 24.4±2.3 years 181.8±5.6 cm 86.4±8.3 kg | ≥1 years | Deadlift (FW; DLw, DLn) | CO | 179.0±29.9 kg (DLw) 162.0±26.9 kg (DLn) |
| Macarilla et al. [27] | n = 17 (M), RT experience | 23.47±4.23 years 175.20±6.09 cm 83.86±13.30 kg | 4.38±1.92 years | Squat (FW) Bench press (FW) | EC (SQ) CO (BP) | 142.94±37.81 kg (SQ) 108.62±23.06 kg (BP) |
| Soriano et al. [53] | n = 27 (16M, 11F), competitive weightlifters | 31.4±6.7 years (M) 29.0±6.3 (F) 178.9±6.2 cm (M) 165.3±4.6 cm (F) 82.8±12.5 kg (M) 60.7 ±4.8 kg (F) | 4.4±5.6 years (M) 2.9±2.5 years (F) | Overhead press (FW) | CO | 57.0±19.0 kg (Overall) |
| Kjær et al. [54] | n = 11 (6M, 5F), elite sprinters | 22.0±2.8 years (M) 22.0±4.1 years (F) 182.7±6.1 cm (M) 167.0±6.0 cm (F) 76.7±6.2 kg (M) 61.0±5.4 kg (F) | ≥2 years | Half-squat (FW) | EC | 190.0±43.1 (RS) 178.2±43.3 (FS) |

**Note:** 1RM = one-repetition maximum; BOR = bent-over-row; BP = bench press; CL = close grip width; CO = concentric-only technique; CSSS = collegiate sports science students; Direct = direct method of predicting one-repetition maximum; DLn = deadlift performed with lifting straps; DLw = deadlift performed without lifting straps; EC = eccentric–concentric technique; F = female; FS = fatigued state; FW = free-weight; LPD = lat pulldown; M = male; Ma = machine; ME = medium grip width; NA = not applicable; NVF = no velocity feedback; RS = rested state; RT = resistance training; SCR = seated cable row; SE = self-selected grip width; SM = Smith machine; SQ = squat; T1 = test 1; T2 = test 2; Overall = data containing all subjects; VF = velocity feedback; WI = wide grip width

**Table 3. Summary of study findings.**

| Author | Device | Velocity variable | Exercise (modality) | Execution Technique | Test load (%1RM) | Predictive 1RM (M ±SD) | Validity | Reliability |
|---|---|---|---|---|---|---|---|---|
| García-Ramos et al. [26] | LPT (T-Force System; Ergotech, Murcia, Spain) | MV MPV | Bench press (SM) | CO EC | CO: MV: 37.8, 75.5 (a); MPV: 40.4, 76.7 (b) EC: MV: 52.1, 82.7 (c); MPV: 55.3, 82.9 (d) | a: 76.3±13.0 kg; b: 75.8±12.8 kg; c: 83.4 ±12.9 kg; d: 81.6 ±12.8 kg | a: $P > 0.05$, ES = 0.02, r = 0.957, SB±RE = -0.2 ±3.7 kg, $R^2$ = 0.053 b: $P > 0.05$, ES = 0.04, r = 0.956, SB±RE = -0.5 ±3.8 kg, $R^2$ = 0.082 c: $P < 0.05$, ES = 0.17, r = 0.976, SB±RE = 2.3 ±3.1 kg, $R^2$ = 0.072 d: $P > 0.05$, ES = 0.03, r = 0.977, SB±RE = 0.4 ±3.0 kg, $R^2$ = 0.080 | RI: in a week a: $P = 0.373$, ES = 0.07, CV = 4.55% (95%CI: 3.60, 6.19), ICC = 0.92 (95%CI: 0.84, 0.96) b: $P = 0.735$, ES = 0.03, CV = 5.11% (95%CI: 4.04, 6.96), ICC = 0.90 (95%CI: 0.79, 0.95) c: $P = 0.429$, ES = 0.05, CV = 3.16% (95%CI: 2.50, 4.30), ICC = 0.95 (95%CI: 0.89, 0.97) d: $P = 0.215$, ES = 0.08, CV = 3.05% (95%CI: 2.41, 4.15), ICC = 0.95 (95%CI: 0.89, 0.98) |
| García-Ramos et al. [43] | LPT (T-Force System; Ergotech, Murcia, Spain) | MV | Prone bench pull (FW) | CO | 48.9, 82.2 | 90.5±13.6 kg | $P = 1.000$, ES = 0.06, r = 0.926, SB±RE = 0.78 ±5.30 kg, $R^2 < 0.001$ | RI: 72–96 hours $P = 0.475$, ES = 0.11, CV = 6.89% (95%CI: 5.17, 10.33), ICC = 0.81 (95%CI: 0.56, 0.93) |
| Pérez-Castilla et al.* [44] | LPT (T-Force System; Ergotech, Murcia, Spain [a]; Chronojump; Barcelona, Spain [b]; Speed4Lift; Madrid, Spain [c]) CBOD (Velowin; DeporTeC, Murcia, Spain [d]) IMU (PUSH band; PUSH Inc., Toronto, Canada [e]; Beast sensor; Beast Technologies Srl., Brescia, Italy [f]) APP (PowerLift; [g]) | MV | Bench press (SM) | CO | 45.0, 85.0 | a: 89.4±13.1 kg; b: 83.8±13.2 kg; c: 88.9 ±12.8 kg; d: 88.1 ±13.6 kg; e: 76.2 ±11.3 kg; f: 90.0 ±15.6 kg; g: 90.1 ±13.3 kg | a: $P = 0.001$, ES = 0.35, r = 0.97 b: $P = 0.372$, ES = 0.08, r = 0.96 c: $P = 0.001$, ES = 0.31, r = 0.97 d: $P = 0.013$, ES = 0.24, r = 0.97 e: $P < 0.001$, ES = 0.70, r = 0.93 f: $P = 0.258$, ES = 0.36, r = 0.50 g: $P = 0.002$, ES = 0.40, r = 0.95 | NA |
| Caven et al. [45] | LPT (GymAware; Kinetic Performance Technology, Canberra, Australia) | MV | Bench press (FW; $GV_{1RM}$, $IV_{1RM}$)Squat (FW; $GV_{1RM}$, $IV_{1RM}$) | EC | BP: 40.0, 90.0 SQ: 20.0, 90.0 | BP: $GV_{1RM}$: 41.3 ±9.0 kg; $IV_{1RM}$: 41.3 ±9.1 kg SQ: $GV_{1RM}$: 90.3±17.8 kg; $IV_{1RM}$: 89.4±18.1 kg | BP: $GV_{1RM}$: $P > 0.05$, ES = 0.32, r = 0.84; $IV_{1RM}$: $P > 0.05$, ES = 0.33, r = 0.89 SQ: $GV_{1RM}$: $P > 0.05$, ES = 0.29, r = 0.76; $IV_{1RM}$: $P > 0.05$, ES = 0.29, r = 0.93 | NA |
| Pérez-Castilla et al.* [46] | LPT (T-Force System; Ergotech, Murcia, Spain) | MV | Bench press (SM; CL, ME, WI and SE) | CO | 46.4, 84.5 | CL: 82.4±12.9 kg ME: 84.5±11.1 kg WI: 80.4±13.2 kg SE: 83.0±13.1 kg | CL: $P = 0.087$, ES = 0.404, r = 0.98, SB ±RE: 3±3 kg, $R^2$ = 0.04 ME: $P < 0.001$, ES = 1.007, r = 0.97, SB ±RE: 2±3 kg, $R^2$ = 0.03 WI: $P = 0.015$, ES = 0.596, r = 0.98, SB ±RE: 1±3 kg, $R^2$ = 0.02 SE: $P = 0.002$, ES = 0.814, r = 0.96, SB ±RE: 2±4 kg, $R^2$ = 0.00 | NA |

(*Continued*)

**Table 3.** (*Continued*)

| Author | Device | Velocity variable | Exercise (modality) | Execution Technique | Test load (%1RM) | Predictive 1RM (M ±SD) | Validity | Reliability |
|---|---|---|---|---|---|---|---|---|
| Fernandes et al.* [47] | Rotary encoder (FitroDyne; Fitronic, Bratislava, Slovakia) | MV | Bench press (SM) Bent-over-row (SM) | EC (BP) CO (BOR) | 20.0, 80.0 (a); 20.0, 40.0 (b); 60.0, 80.0 (c) | NA | Whole: BP: a: $P = 0.531$, ES = 0.05, r = 0.87; b: $P = 0.220$, ES = 0.21, r = 0.58; c: $P = 0.324$, ES = 0.08, r = 0.87. BOR: a: $P < 0.001$, ES = 0.56, r = 0.77; b: $P = 0.403$, ES = 0.14, r = 0.72; c: $P = 0.002$, ES = 0.50, r = 0.77 Young: BP: a: $P = 0.743$, ES = 0.05, r = 0.80; b: $P = 0.870$, ES = 0.03, r = 0.63; c: $P = 0.902$, ES = 0.02, r = 0.81. BOR: a: $P = 0.014$, ES = 0.54, r = 0.71; b: $P = 0.964$, ES = 0.01, r = 0.68; c: $P = 0.023$, ES = 0.52, r = 0.72 Middle-aged: BP: a: $P = 0.098$, ES = 0.17, r = 0.90; b: $P = 0.069$, ES = 0.56, r = 0.61; c: $P = 0.056$, ES = 0.20, r = 0.90. BOR: a: $P = 0.004$, ES = 0.69, r = 0.70; b: $P = 0.080$, ES = 0.56, r = 0.55; c: $P = 0.035$, ES = 0.50, r = 0.81 | NA |
| Janicijevic et al.* [48] | LPT (T-Force System; Ergotech, Murcia, Spain) | MV | Bench press (SM; $GV_{1RM}$ [a], $IV_{1RM}$ [b], $IV_{last}$ [c]) | CO EC | 45.0, 90.0 | CO: a: 64.2±17.6 kg; b: 64.2±17.7 kg; c: 64.7±17.7 kg EC a: 64.9±17.8 kg; b: 64.7±17.7 kg; c: 64.7±17.7 kg | CO: a: ES = 0.08, r = 0.99; b: ES = 0.08, r = 0.99; c: ES = 0.12, r = 0.99 EC a: ES = 0.14, r = 0.98; b: ES = 0.14, r = 0.98; c: ES = 0.16, r = 0.98 | NA |
| Perez et al. [49] | LPT (GymAware; Kinetic Performance Technology, Canberra, Australia) | MV | Bench press (FW) | EC | 50.0, 90.0 (a); 70.0, 90.0 (b); 50.0, 70.0 (c) | Overall: a: 90.99 ±46.00 kg; b: 91.29 ±45.61 kg; c: 93.19 ±47.35 kg | a: $P = 0.67$, ES = 0.02, r = 0.99, SB±RE: 0.76 ±7.93 kg b: $P = 0.61$, ES = 0.02, r = 0.98, SB ±RE: 1.06±9.05 kg c: $P = 0.12$, ES = 0.06, r = 0.99, SB±RE 2.96 ±8.57 kg | NA |
| Pérez-Castilla et al.[50] | LPT (Real Power Pro Globus; Codogne, Italy) APP (PowerLift) | MV | Lat pulldown (Ma) Seated cable row (Ma) | CO | 40.0, 85.0 | Overall LPT: LPD: 66.1±19.6 kg; SCR: 67.0±19.6 kg APP: LPD: 62.9±18.8 kg; SCR: 64.5±18.8 kg | LPT: LPD: $P > 0.05$, ES = 0.04, r = 0.98, SB ±RE = 0.80±4.29 kg, $R^2 = 0.00$; SCR: $P > 0.05$, ES = 0.00, r = 0.99, SB ±RE = 0.02±3.79 kg, $R^2 = 0.01$ APP: LPD: $P > 0.05$, ES = 0.08, r = 0.98, SB±RE = 1.75 ±5.42 kg, $R^2 = 0.30$; SCR: $P > 0.05$, ES = 0.06, r = 0.96, SB ±RE = -1.11±5.36 kg, $R^2 = 0.07$ | NA |

(*Continued*)

**Table 3.** (*Continued*)

| Author | Device | Velocity variable | Exercise (modality) | Execution Technique | Test load (%1RM) | Predictive 1RM (M ±SD) | Validity | Reliability |
|---|---|---|---|---|---|---|---|---|
| Aidar et al. *[51] | LPT (Speed4Lift; Madrid, Spain) | MV | Bench press (FW; MVT, F-V) | CO | 40.0, 80.0 (a); 50.0, 80.0 (b) | MVT: a: 122.0±41.3 kg; b: 129.4±51.2 kg F-V: a: 144.0±59.7 kg; b: 150.9±64.9 kg | MVT: a: ES = 0.25, b: ES = 0.39; F-V: a: ES = 0.65, b: ES = 0.74 | NA |
| Çetin et al. [52] | LPT (GymAware; Kinetic Performance Technology, Canberra, Australia) | MV | Deadlift (FW) Half-squat (FW) | CO | 40.0, 60.0 (a); 40.0, 80.0 (b); 40.0, 90.0 (c); 60.0, 80.0 (d); 60.0, 90.0 (e) | Deadlift: a: 218.96 ±83.41 kg; b: 182.94 ±33.17 kg; c: 173.46 ±29.38 kg; d: 179.15 ±25.59 kg; e: 174.40 ±32.24 kg Half-squat: a: 183.86 ±61.54 kg; b: 171.11 ±40.52 kg; c: 155.35 ±39.02 kg; d: 189.87 ±102.06 kg; e: 153.85±39.77 kg | Deadlift: a: $P = 0.0305$, ES = 0.74, r = 0.861; b: $P = 0.9830$, ES = 0.34, r = 0.903; c: $P > 0.9999$, ES = 0.06, r = 0.972; d: $P = 0.9981$, ES = 0.25, r = 0.856; e: $P > 0.9999$, ES = 0.08, r = 0.956 Half-squat: a: $P = 0.5535$, ES = 0.70, r = 0.000; b: $P = 0.8889$, ES = 0.59, r = 0.610; c: $P = 0.9992$, ES = 0.18, r = 0.840; d: $P = 0.2874$, ES = 0.54, r = 0.055; e: $P = 0.9998$, ES = 0.14, r = 0.872 | RI: 48 hours Deadlift: a: CV = 7.72%, ICC = 0.171; b: CV = 4.23%, ICC = 0.815; c: CV = 4.27%, ICC = 0.996; d: CV = 3.86%, ICC = 0.335; e: CV = 4.62%, ICC = 0.972 Half-squat: a: CV = 6.26%, ICC = 0.235; b: CV = 4.07%, ICC = 0.822; c: CV = 4.34%, ICC = 0.905; d: CV = 9.81%, ICC = 0.479; e: CV = 4.50%, ICC = 0.988 |
| Jiménez-Alonso et al.*[53] | LPT (T-Force System; Ergotech, Murcia, Spain) | MV | Bench press (FW; VF, NVF) | EC | 40.0, 85.0 | VF: 82.6±18.4 kg; NVF: 83.5±17.4 kg | VF: ES = 0.17, r = 0.99, SB±RE = 3.20±2.61 kg, $R^2 = 0.09$ NVF: ES = 0.09, r = 0.97, SB ±RE = 1.47±4.28kg, $R^2 = 0.02$ | NA |
| Jukic et al. [54] | LPT (GymAware; Kinetic Performance Technology, Canberra, Australia) | MV | Deadlift (FW; DLw, DLn; $GV_{1RM}$, $IV_{1RM}$) | CO | 40.0, 90.0 | DLw: $GV_{1RM}$: 191.7 ±31.3 kg; $IV_{1RM}$: 195.0±33.1 kg DLn: $GV_{1RM}$: 170.1±28.2 kg; $IV_{1RM}$: 168.6 ±27.4 kg | DLw: $GV_{1RM}$: $P = 0.001$, ES = 0.36, r = 0.93, SB±RE = 10.9 ±11.6kg; $IV_{1RM}$: $P = 0.005$, ES = 0.40, r = 0.89, SB±RE = 13.1 ±17.0 kg DLn: $GV_{1RM}$: $P = 0.235$, ES = 0.12, r = 0.92, SB±RE = 3.3 ±11.4 kg; $IV_{1RM}$: $P = 0.086$, ES = 0.11, r = 0.98, SB±RE = 3.4 ±7.9 kg | NA |
| Macarilla et al.[28] | LPT (The Open Barbell System Version 3; Squats & Science, New York, USA) | MV | Squat (FW) Bench press (FW) | EC (SQ) CO (BP) | 19.88±0.10, 86.69±0.03 (SQ) 21.63 ±0.08, 73.75±0.09 (BP) | SQ: 171.62±36.61 kg; BP: 119.00 ±27.66 kg | SQ: $P < 0.05$, ES = 1.89, r = 0.938, SB ±RE = 29.1176±13.7177 kg BP: $P < 0.05$, ES = 0.83, r = 0.933, SB ±RE = 9.7500±10.3859 kg | NA |
| Soriano et al.*[55] | LPT (Chronojump; Barcelona, Spain) | MV | Overhead press (FW) | CO | 45.0, 75.0 (a) 45.0, 90.0 (b) | a: 52.1±19.5 kg b: 53.9±19.1 kg | a: $P = 0.296$, ES = 0.07, r = 0.941, SB±RE = -1.4 ±6.6 kg, $R^2 = 0.002$ b: $P = 0.164$, ES = 0.05, r = 0.982, SB±RE = -1.0 ±3.6 kg, $R^2 = 0.002$ | NA |

(*Continued*)

**Table 3.** (Continued)

| Author | Device | Velocity variable | Exercise (modality) | Execution Technique | Test load (%1RM) | Predictive 1RM (M ±SD) | Validity | Reliability |
|---|---|---|---|---|---|---|---|---|
| Kjær et al. [56] | LPT (GymAware; Kinetic Performance Technology, Canberra, Australia) | MV PV | Half-squat (FW) | EC | 50.0, 80.0 | MV: RS: 197.9±49.3 kg; FS: 179.3±45.9 kg PV: RS: 199.1 ±49.8 kg; FS: 192.0 ±56.2 kg | MV: RS: $P$ = 0.249, ES = 0.17; FS: $P$ = 0.808, ES = 0.02 PV: RS: $P$ = 0.207, ES = 0.20; FS: $P$ = 0.181, ES = 0.28 | NA |

**Note:** %1RM = percentage of one-repetition maximum; 95%CI = 95% confidence interval; APP = smartphone application; BOR = bent-over-row; BP = bench press; CBOD = camera-based optoelectronic device; CL = close grip width; CO = concentric-only technique; CV = coefficient of variance; DLn = deadlift performed with lifting straps; DLw = deadlift performed without lifting straps; EC = eccentric–concentric technique; ES = effect size; FS = fatigued state; F-V = Predicting 1RM by force-velocity relationship method; FW = free-weight; $GV_{1RM}$ = group velocity value of one-repetition maximum; ICC = interclass correlation coefficient; IMU = inertial measurement units; $IV_{1RM}$ = individual velocity value of one-repetition maximum; $IV_{last}$ = individual velocity value of last repetition of a set; LPD = lat pulldown; LPT = linear position transducers; Ma = machine; ME = medium grip width; MPV = mean propulsive velocity; MV = mean velocity; MVT = Predicting 1RM by minimal velocity threshold method; NA = not applicable; NVF = no velocity feedback; $P$ = $P$-value; RI = retest interval; RS = rested state; $R^2$ = coefficient of determination; SB±RE = systematic bias ± random error; SCR = seated cable row; SE = self-selected grip width; SM = smith machine; SQ = squat; Overall = data containing all subjects; VF = velocity feedback; WI = wide grip width

\* Using the velocity of 1RM from previous research

## Discussion

The primary objective of this systematic review was to assess the reliability and validity of the two-point method in predicting 1RM compared to the direct method. Additionally, we aimed to identify factors that could potentially influence the accuracy of the two-point method in 1RM prediction. In the subsequent discussion, we presented an analysis of the validity and reliability of the two-point method for predicting 1RM. Furthermore, based on the extracted data, we examined three factors that may impact the accuracy of the two-point method, including the choice of exercises, velocity measurement devices, and selection of test loads, and other potential factors that could affect the accuracy of the two-point method. Lastly, we offered practical recommendations for future research on the two-point method, aiming to advance the knowledge and understanding in this field.

### Validity of two-point method

Out of the 16 studies included in this review, a total of 58 1RM values predicted by the two-point method were obtained. One study did not provide the necessary data for calculating the 1RM values measured by the direct method and predicted by the two-point method [45]. Among the included studies, a total of 73 ES values were obtained. Out of these, 40 (54.8%) ES had values less than 0.2 (trivial), 30 (41.1%) ES ranged from 0.2 to 0.59 (small), two (2.7%) ES ranged from 0.6 to 1.2 (moderate), and one (1.4%) ES fell between 1.2 and 2.0 (large).

Upon conducting quantitative synthesis of the 58 data points, the overall ES for 1RM predicted by the two-point method was found to be 0.203 (small) (95%CI: 0.132, 0.275; $P < 0.001$). This indicates that although the 1RM predicted by the two-point method was significantly higher than the actual 1RM, the difference was small in magnitude (Fig 2). Although the overall ES was not within our pre-defined acceptable range (less than 0.2), it was very close.

In addition, we extracted 71 r values from the included studies, there were 45 (63.4%) range from 0.90 to 0.99 (nearly perfect), 17 (23.9%) range from 0.70 to 0.89 (very large), 7 (9.9%) range from 0.50 to 0.69 (large) and two (2.8%) less than 0.09 (trivial). The r value represents the degree of correlation between the mean of predicted 1RM by two-point method and actual 1RM, which means that the closer the r value to 1, the higher the validity of the two-point

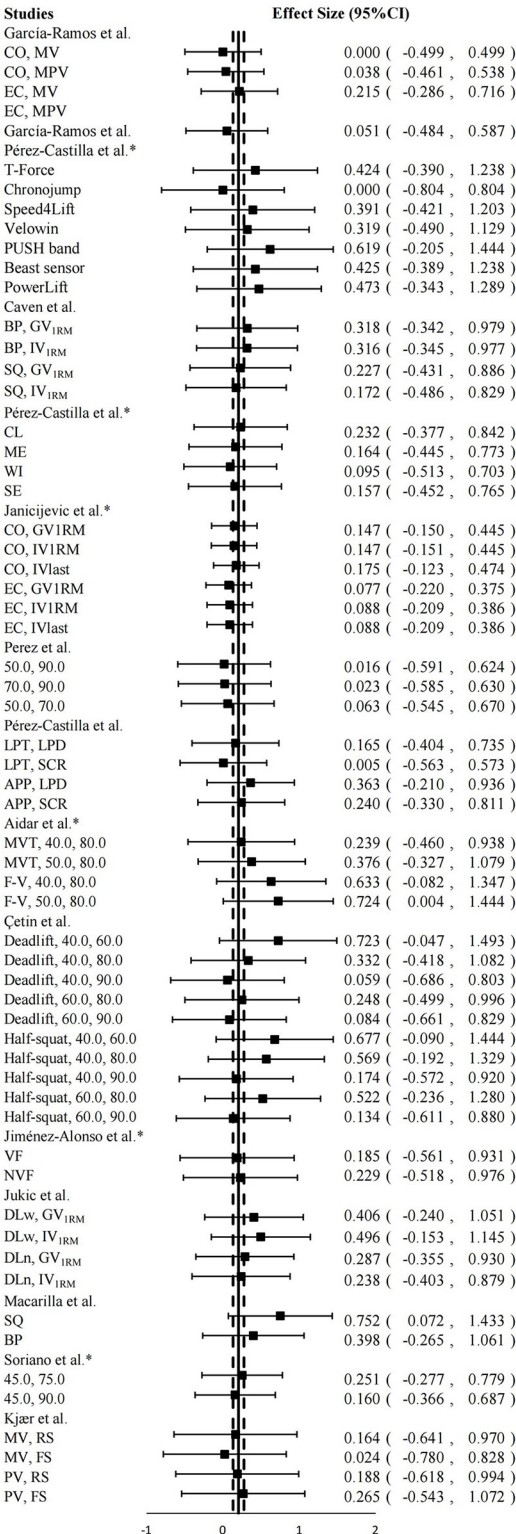

**Fig 2. Mean (■) effect size (95%CI), overall composite effect size (—), 95%CI of overall composite effect size (- ·—· -) for magnitude of difference between predicted 1RM by two-point method and 1RM value by direct method.** It should be noted that the smaller the ES, the more accurate the two-point method predicts 1RM.

method. We found that the lower r value mainly came from the poor accuracy of the velocity measurement device (IMU) or unreasonable selection of test loads (20%1RM difference) [42, 56]. In general, the 1RM predicted by the two-point method was highly correlated with the actual 1RM in most of the included studies.

Therefore, our findings demonstrated that the two-point method was a valid approach for predicting 1RM based on current evidence.

## Reliability of two-point method

Out of the 16 included studies, only three examined the reliability of the two-point method in predicting 1RM [19, 26, 50], resulting in a total of 15 generated ICC values. Among these, eight (53.3%) ICC values were classified as excellent (ICC > 0.90), three (20.0%) were classified as good (ICC = 0.75–0.90), and four (26.7%) were classified as poor (ICC < 0.5). The overall ICC value for the two-point method was approximately 0.797, calculated using statistical methods. This indicated that the overall reliability of the two-point method was good (0.75–0.90), but not excellent. Notably, we observed that the "poor" ICC values were exclusively reported by Çetin et al. [50], and the specific reasons behind the generation of these "poor" values could be determined that the 2 test loads were very close (20%1RM difference). As recently reported by García Ramos [56], the velocity at which the same subject performed the same exercise with same intensity was not fixed, but inevitably fluctuated. The smaller the difference between the two test loads, the greater the impact on LVR, which also led to a decrease in the reliability of the two-point method in predicting 1RM. When the unreasonable load (20% 1RM difference) was eliminated, the calculated overall ICC was excellent (ICC = 0.911). Therefore, RT practitioners should pay attention to the selection of reasonable test loads to obtain reliable results when applying the two-point method to predict 1RM.

## Factors that may affecting the accuracy of two-point method

**Exercises selection.**   In the included studies, we observed that the choice of exercises had an impact on the accuracy of the two-point method in predicting 1RM. Overall, the two-point method demonstrated greater accuracy in predicting 1RM for upper-limb exercises (ES = 0.172 [95%CI: 0.091, 0.252], $P < 0.001$, trivial) compared to lower-limb exercises (ES = 0.325 [95%CI: 0.167, 0.483], $P < 0.001$, small) (Fig 3). This difference may be attributed to the fact that lower-limb exercises, such as free-weight squat and deadlift, involve a greater number of joints and muscles and require more complex technique. Particularly when individuals have poor motor skills, the velocity measurement device may overestimate the velocity value more significantly in lower-limb exercises compared to upper-limb exercises due to the asymmetrical anterior-posterior and medial-lateral horizontal movement of the barbell [32, 57]. However, in studies where a large discrepancy between predicted 1RM and actual 1RM was observed [27, 50], the authors did not provide further specific information regarding the subjects' "exercise techniques", such as video analysis of barbell movement trajectories. Therefore, it remains unclear whether these less accurate data were a result of issues with the experimental design or if the subjects' "poor exercise techniques" influenced the outcomes. In summary, RT practitioners should be aware that this may produce greater errors than upper-limb exercises when applying the two-point method to predict 1RM of lower-limb exercises.

In theory, the use of a Smith machine during exercise can constrain the barbell's movement to the vertical direction, which can mitigate the velocity error caused by the barbell's asymmetrical anterior-posterior and medial-lateral horizontal movement. This feature may enhance the accuracy of 1RM prediction based on LVR, especially for individuals who are new to RT. However, from practical experience, athletes prefer to use free-weight for RT rather than

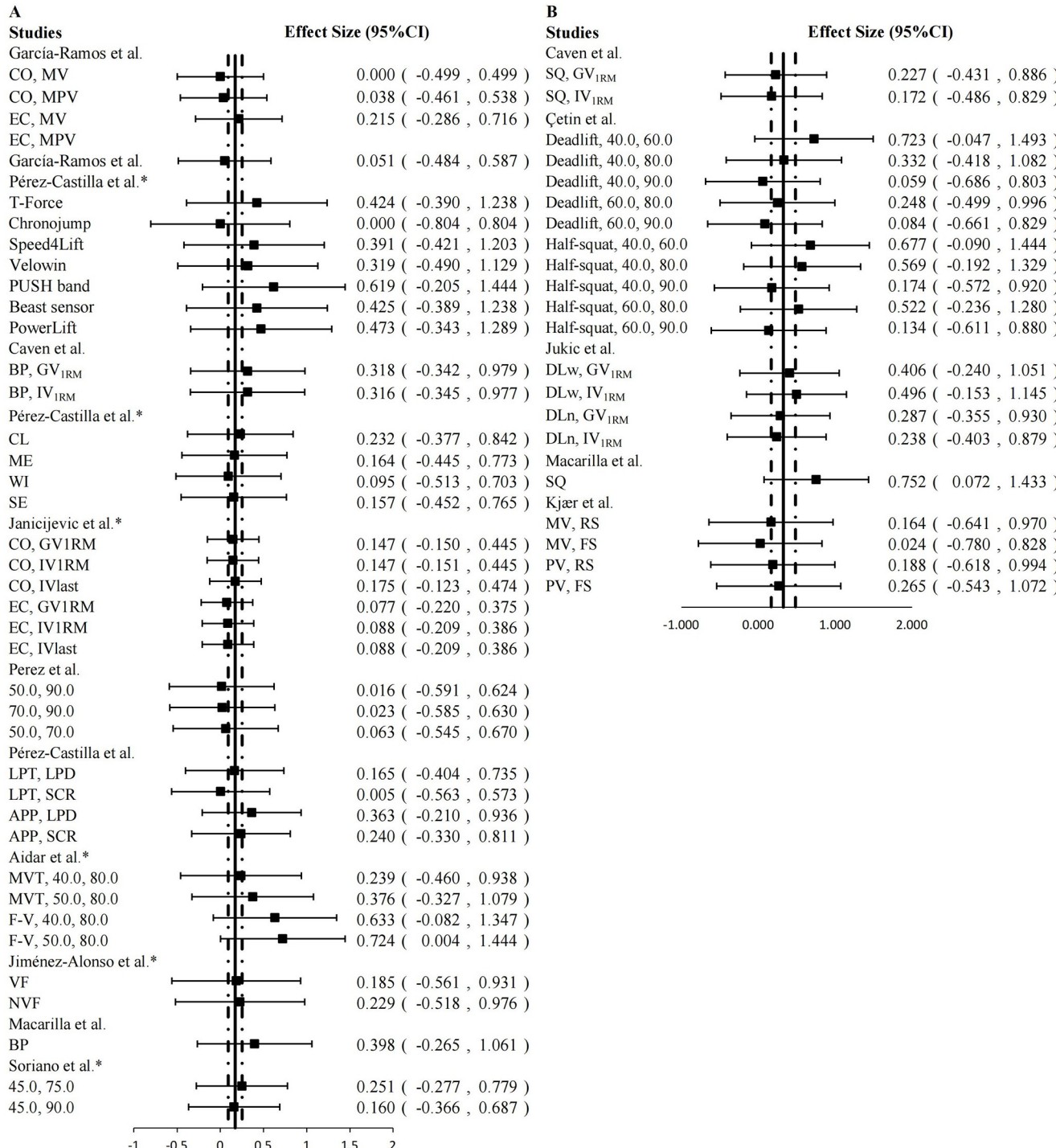

**Fig 3. Mean (■) effect size (95%CI), overall composite effect size (—), 95%CI of overall composite effect size (- ·—· -) for magnitude of difference between predicted 1RM by two-point method and 1RM value by direct method.** A: upper-limb exercise; B: lower-limb exercise. It should be noted that the smaller the ES, the more accurate the two-point method predicts 1RM.

Smith machine. Nevertheless, no study has directly compared the use of different exercise modalities to determine whether they affect the accuracy of the two-point method in predicting 1RM. Currently, a substantial amount of research on the two-point method has been

conducted using the Smith machine and other machines, particularly for upper-limb exercises. Strength and conditioning coaches should be aware when applying these conclusions, as whether these findings can be extrapolated to free-weight exercises warrants further investigation and validation.

Two studies compared the effects of different execution techniques on the accuracy of the two-point method in predicting bench press 1RM [26, 46]. It was observed that the use of the eccentric-concentric (EC) technique, which involves performing the concentric contraction immediately after the eccentric contraction, slightly diminished the accuracy of the two-point method in predicting 1RM compared to the concentric-only (CO) technique, where a pause is introduced after the eccentric contraction [26, 46]. However, its accuracy was still within the acceptable range (ES < 0.2). Therefore, RT practitioners can use different execution techniques when applying the two-point method at will, as it does not significantly affect the accuracy of the two-point method in predicting 1RM.

Two of the included studies investigated different variations of the exercises, specifically bench press with different grip widths and deadlift with or without lifting straps [44, 52]. Interestingly, the accuracy of the two-point method in predicting 1RM for bench press did not seem to be significantly influenced by different grip widths, as the ES ranged from 0.10 to 0.22. However, when it came to deadlift, the use of lifting straps resulted in a moderate overestimation of 1RM compared to performing the exercise without lifting straps, with ES ranging from 0.36 to 0.40 for lifting straps and 0.11 to 0.12 for no lifting straps. Regarding the impact of different exercise variants on the prediction of 1RM using LVR, some researchers believed that the same LVR can be extrapolated to different exercise variants [14, 58], while others held different opinions [59–61]. Whether these findings can be extrapolated from multiple-point LVR to two-point LVR needs further investigation, and RT practitioners need to take these factors into account when applying two-point LVR to predict 1RM for different motor variants. It was worth noting that the use of lifting straps appears to be an exception, as one study [52] reported lower accuracy in predicting 1RM when lifting straps were employed. Therefore, coaches, athletes, and researchers should consider whether similar findings may apply to exercises such as barbell rows, pull-ups, and other exercises where lifting straps are utilized.

**Velocity measurement devices selection.** The ES of different types of velocity measurement devices when using the two-point method to predict 1RM are presented in Fig 4. However, the results obtained from the Rotary encoder were not included in the data synthesis due to the unavailability of the original data pertaining to predicted 1RM and actual 1RM in the corresponding article.

As depicted in Fig 4, the synthesized ES of LPT, CBOD, IMU and APP were 0.192 (95%CI: 0.118, 0.266; $P < 0.001$), 0.240 (95%CI: -1.080, 0.590; $P < 0.013$), 0.521 (95%CI: -0.059, 1.100; $P = 0.078$) and 0.335 (95%CI: -0.027, 0.698; $P = 0.069$), respectively.

LPT is one of the most common movement velocity monitoring devices in RT, and generally shows the highest accuracy compared to other movement velocity devices. Therefore, some researchers have called LPT together with 3d motion capture devices as the "gold standard" for monitoring movement velocity in RT [62, 63]. LPT also showed the highest accuracy in this review (ES = 0.192). Although APP showed similar accuracy to LPT in the included studies, it should be noted that the studies only used the APP for velocity measurements in exercises performed on Smith machines and machines with fixed movement trajectories [42, 48]. However, in free-weight exercises where the barbell's trajectory is more complex, using an APP that measures velocity through a 2D plane may introduce larger errors compared to LPT and 3D motion capture devices [64].

Compared to other devices, the IMU exhibited the worst accuracy in this systematic review. This discrepancy does not meet our predetermined criteria. Orser et al. [65] suggested that the

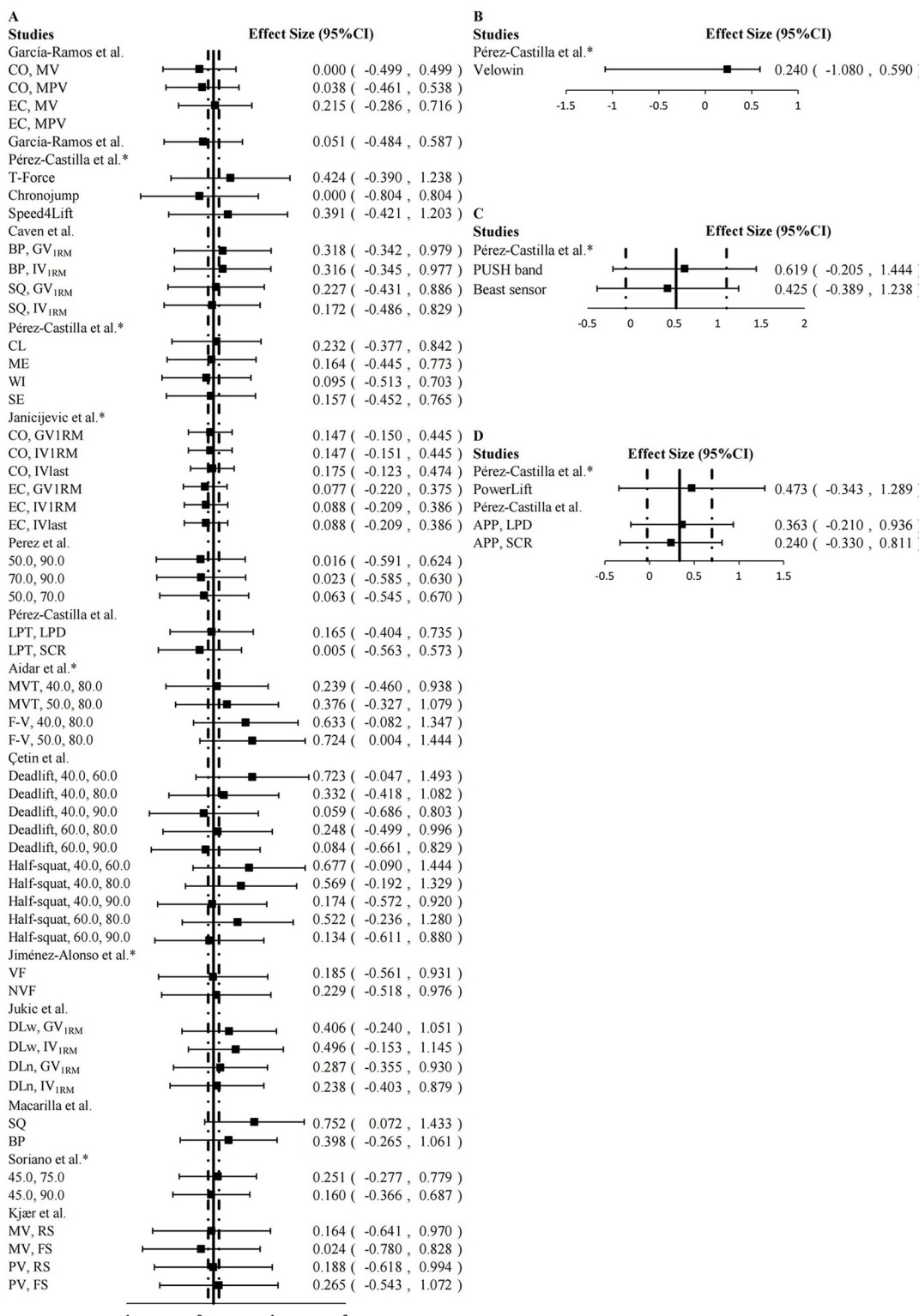

**Fig 4. Mean (■) effect size (95%CI), overall composite effect size (—), 95%CI of overall composite effect size (- ·—· -) for magnitude of difference between predicted 1RM by two-point method and 1RM value by direct method.** A: linear position transducers (LPT); B: camera-based optoelectronic device (CBOD); C: inertial measurement units (IMU); D: smartphone application (APP). It should be noted that the smaller the ES, the more accurate the two-point method predicts 1RM.

IMU could accurately measure velocity values during low-velocity motions but not during high-velocity motions. This contradicts the conditions required for the application of the two-point method, which necessitates the selection of both lighter and heavier loads to accurately predict 1RM [32]. Pérez-Castilla et al. [42] also concluded that the IMU was not suitable for the two-point method in 1RM prediction because it couldn't accurately measure movement velocity.

Regarding the FitroDyne rotary encoder, Fernandes et al. reported an ES range of -0.56 to 0.69 [45]. Furthermore, they also noted that FitroDyne (rotary encoder) and GymAware (LPT) could not record peak or mean velocity with acceptable agreement, and their data should not be used interchangeably or compared [66].

In conclusion, if aiming to obtain a more accurate 1RM value using the two-point method, the preferred choice for velocity measurement equipment would be the use of LPT.

**Teat loads selection.**   Twenty-one different combinations of test loads were used to develop the two-point method model in the included studies. Most of these studies utilized a lighter load ($\leq$ 50% 1RM) and a heavier load ($>$ 50% 1RM) as the test loads for the two-point method. It was observed that the greater the difference between the two loads, the more accurate the predicted 1RM tended to be, as recently reported by García Ramos [56]. For instance, Çetin et al. [50] reported an ES of 0.74 for deadlift when using 40.0% and 60.0% 1RM as test loads, whereas the ES was 0.06 when using 40.0% and 90.0% 1RM as test loads. Among all the test load combinations, the largest differences were found to be 20.0% and 90.0% [43].

However, it should be noted that all studies included in this review employed an incremental load test, followed by the selection of two loads and their corresponding velocity values for the two-point method. This approach differs from actual RT, as the muscle's physiological function improves during incremental load testing, whereas the two-point method in RT practice strictly tests two loads to predict 1RM. Consequently, the conclusions drawn from the literature may not directly apply to RT practice, as the impact of additional warm-up sets on the test results remains unclear. Ribeiro et al. [67] reported a significant decrease in movement velocity during heavy load (80% 1RM) without specific warm-up, while Miras-Moreno et al. [68] reported that additional incremental specific warm-up compromised the magnitude of the LVR variables. It is also uncertain whether a larger difference between the two loads leads to more accurate prediction results or if there exists an optimal interval for maximizing the accuracy of predicted 1RM.

To ensure better applicability of relevant research findings to RT practice, future studies in the field of the two-point method should strictly adhere to using only two loads for testing in each experiment. At present, some studies have made some efforts [68–71], but it seems that there was no study involving 1RM prediction, which is worthy of research.

**Other factors.**   *Movement velocity variable.* Movement velocity variables play a crucial role in velocity-based methods used in RT. The three primary velocity variables of interest are MV, MPV and PV, each with its own definition. MV represents the average velocity from the beginning of the concentric phase until the load reaches its maximum height [29]. MPV refers to the average velocity from the start of the concentric phase until the load's acceleration becomes lower than gravity (-9.81 m/s$^2$) [55]. PV represents the maximum instantaneous velocity achieved from the start of the concentric phase until the load reaches its maximum height [29].

Previous studies have indicated that MV and MPV are more suitable velocity variables than PV when defining LVR in non-ballistic exercises [72–74]. However, in ballistic exercises, MPV may not be appropriate due to the limitations of current commercial velocity measurement devices in accurately calculating takeoff time. Therefore, MV and PV are considered more appropriate in such cases [75, 76]. It is worth noting that among the 16 studies included in this systematic review, only one analyzed MPV [26] and PV [54], while the remaining studies

focused solely on MV. Furthermore, this systematic review did not cover ballistic exercises. Consequently, further research is needed to determine whether the choice of different velocity variables impacts the accuracy of 1RM prediction using the two-point method.

*1RM determination method*. This systematic review included four methods of determining 1RM: direct testing of $V_{1RM}$, obtaining $V_{1RM}$ values from previous studies, the $V_{last}$ method (using the velocity of the last repetition in a specific load set as $V_{1RM}$), and the F-V method (calculating 1RM value based on the intersection of the force-velocity relationship and the weight-velocity relationship).

Theoretically, directly testing $V_{1RM}$ for each subject was considered the most accurate approach, as previous research has demonstrated substantial between-subject CV in $V_{1RM}$ [29, 77, 78]. Using the same $V_{1RM}$ value for all subjects would lead to inaccurate predictions of 1RM using the two-point method.

In practice, when testing athletes' 1RM using the two-point method, coaches often rely on $V_{1RM}$ values reported in previous studies to save time. However, it is important to consider whether the conditions in those studies align with the athletes' specific circumstances, such as training experience, strength level, height, age and other relevant factors. This places certain requirements on the coaches' expertise.

The $V_{last}$ method appeared to be effective for upper-limb exercises [43]. However, in lower-limb exercises, Lake et al. [79] reported significantly lower predicted deadlift 1RM using the $V_{last}$ method compared to actual deadlift 1RM (P < 0.05, ES = 1.03–1.75). Therefore, when using the two-point method to predict 1RM for lower-limb exercises, it is advisable to avoid using the $V_{last}$ method to determine the $V_{1RM}$ value.

The F-V method demonstrated a moderate error (ES = 0.65–0.74) in predicting 1RM. Picerno et al. [80] suggested that the F-V method required a minimum of three loads to accurately predict 1RM. The use of the two-point method inevitably influences the accuracy of the weight-velocity relationship and force-velocity relationship, thus affecting the location of the intersection of these two curves. Therefore, the F-V method is not suitable for the two-point method and its application in this context should be avoided.

*Velocity feedback*. In this systematic review, a study examined the effect of providing velocity feedback on the accuracy of 1RM prediction using the two-point method, comparing the use of velocity feedback with and without it [51]. The results showed that although the error in predicted 1RM values with velocity feedback was slightly higher compared to without feedback (ES = 0.17 and 0.09, respectively), this difference was trivial (ES < 0.2).

One of the primary objectives of RT is to maximize strength levels and power output. Research has consistently demonstrated that providing velocity feedback during training can lead to greater improvements in strength and performance. Therefore, when utilizing the two-point method to predict 1RM, it is advisable to incorporate velocity feedback as well. This will enhance the effectiveness of RT and better align with the overarching goals of strength and power development.

*Fatigued state*. In this systematic review, one study compared the effect of fatigue on the accuracy of the two-point method for predicting 1RM [54]. The results showed that that 1RM predicted by the two-point method with different velocity variable (MV or PV) was not significantly different from the actual 1RM under fatigue or non-fatigue conditions. Moreover, when MV variable was used, the 1RM predicted by two-point method was even more accurate under fatigue condition than under non-fatigue condition (ES = 0.02 and ES = 0.17, respectively).

For athletes, because of the need for a large number of specific trainings, RT was usually not in a completely fatigue-free state. Therefore, this study was very important for athletes, it showed that whether in a certain state of fatigue or not can be applied to the two-point method

of maximum strength test. However, further research is needed to determine whether this conclusion can be extended to exercises other than the half-squat.

## Conclusion

This systematic review supported the validity of the two-point method for predicting 1RM, also its reliability as long as the test loads were chosen reasonably (Large difference between two test loads). Several factors were identified to affect the accuracy of 1RM prediction, such as the choice of exercise, velocity measurement device and test load. Specifically, upper-limb exercises demonstrated higher accuracy compared to lower-limb exercises, and linear position transducers exhibited superior performance among the different measurement devices. Theoretically, the greater the difference between the two test loads, the higher the accuracy of predicting 1RM. However, there was no study to explore the accuracy of two-point method in 1RM prediction based on only testing two loads. Other factors that may influence accuracy include the selection of velocity variable, 1RM determination method, the provision of velocity feedback and the state of fatigue. Although this systematic review attempts to clarify the factors that influence the accuracy of the two-point method in predicting 1RM, there are some limitations. When analyzing one factor, more factors will inevitably be involved. For example, when analyzing the data of speed measuring equipment, it is inevitable that both upper and lower limb movements will be considered at the same time. But overall, this systematic review will provide useful guidance for the follow-up study and promote the further application of the two-point method in RT.

### Practical applications

Based on the results of this study, it is feasible to apply the two-point method to predict 1RM in RT practice, but the following issues need to be noted: 1) Need to use high accuracy velocity measurement devices, such as LPT; 2) Select a suitable 1RM determination method (appropriate $V_{1RM}$); 3) Select two test loads with large difference; 4) The accuracy of upper-limb exercises are higher than that of lower-limb exercises.

## Supporting information

**S1 Checklist.**
(DOCX)

**S1 Data.**
(XLSX)

**S1 File.**
(PDF)

## Acknowledgments

Thanks to all the authors who contributed to this study.

## Author Contributions

**Conceptualization:** Zongwei Chen, Xiuli Zhang.

**Data curation:** Zongwei Chen, Zheng Gong, Liwen Pan.

**Formal analysis:** Zongwei Chen, Zheng Gong, Liwen Pan, Xiuli Zhang.

**Methodology:** Zongwei Chen, Zheng Gong, Xiuli Zhang.

**Supervision:** Zongwei Chen, Xiuli Zhang.

**Writing – original draft:** Zongwei Chen, Xiuli Zhang.

**Writing – review & editing:** Zongwei Chen, Zheng Gong, Xiuli Zhang.

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
