## [Decision Letter · Decision Letter 0]

3 Sep 2023

PONE-D-23-21953Is two-point method a valid and reliable method to predict 1RM? A systematic reviewPLOS ONE

Dear Dr. Zhang,

Thank you for submitting your manuscript to PLOS ONE. After careful consideration, we feel that it has merit but does not fully meet PLOS ONE’s publication criteria as it currently stands. Therefore, we invite you to submit a revised version of the manuscript that addresses the points raised during the review process.

ACADEMIC EDITOR:Two experts have provided an extensive review of your paper. Please pay special attention to the comments related to the statistical analysis because both reviewers made several observations related to this subsection. 

We look forward to receiving your revised manuscript.

Kind regards,

Danica Janicijevic, Ph.D

Academic Editor

PLOS ONE

Reviewers' comments:

Reviewer's Responses to Questions

**Comments to the Author**

1. Is the manuscript technically sound, and do the data support the conclusions?

Reviewer #1: Partly

Reviewer #2: Yes

2. Has the statistical analysis been performed appropriately and rigorously? 

Reviewer #1: Yes

Reviewer #2: Yes

3. Have the authors made all data underlying the findings in their manuscript fully available?

Reviewer #1: Yes

Reviewer #2: Yes

4. Is the manuscript presented in an intelligible fashion and written in standard English?

Reviewer #1: Yes

Reviewer #2: Yes

5. Review Comments to the Author

Reviewer #1: General comment

First, I would like to congratulate the authors for their efforts on the manuscript “Is two-point method a valid and reliable method to predict 1RM? A systematic review”. The article addresses the prediction of the maximum load that can be lifted once with a proficient technique (1RM) using the load-velocity relationship.

The topic is particularly important because in recent years, velocity-based training has gained popularity among strength and conditioning coaches. However, methodological considerations for implementing the two-point load-velocity method to assess 1RM are scarce. In the first part of this systematic review, the reliability and validity of 1RM predicted with load-velocity relationship are analysed. In the second part, the authors provide practical advice on how to perform the two-point method to obtain the most reliable and valid 1RM by explaining the factors that most affect the accuracy of the method. The authors conclude that the two-point load-velocity method is reliable and valid for obtaining the 1RM, outlining the selection of the appropriate velocity and regression models as the most important factors.

One of the most important considerations of this review is the statistical method used to assess the overall validity of the two-point method from the included studies. For each study separately, the authors calculated the difference between the predicted and direct 1RM as the effect size (ES). Based on the average value from all studies, the synthesized validity of the method was assessed. However, using the average values from the studies that had overestimates and under-estimations could lead to an erroneous bias. I suggest this should be corrected as the result could change the interpretation and conclusions. Detailed comments with minor considerations and suggestions can be found as comments in the text below.  

Introduction

The introduction is well written. The background of the evaluation of the 1RM using the load-velocity relationship is clearly described. In addition, the authors provide sound objectives and hypotheses.

Materials and methods

The following are some minor comments.

Comment 1: Lines 92 to 95 indicate the search strategy that led to 21, 28, 29, and 9 sources at the identification level. This is a relatively small number of studies for the first level of search and the main consideration is that important literature was omitted due to the very specific terms used. Please describe why you chose to use such a search strategy.

Comment 2: Under line 113 the fifth (5)) Selection criteria were studies that used the two-point method to predict 1RM, with the exception of L0 and F0. I recommend that the criteria used to exclude the data (lines 114 to 116) be transferred to the Data Extraction section.

Comment 3: When describing F0 (load at zero velocity) in line 114, it is better to use force at zero velocity.

Comment 4: The ES of the differences between the predicted and actual 1RM were calculated. It is not clear from the text (lines 135 to 138) exactly how this was done. Please provide the equation for calculating the ES and include the references for this statistical procedure.

Comment 5: You mention the total ES later in the text of the manuscript. However, it is not clear how this total effect was calculated using the methods. Please describe this.

Comment 6: Why did you choose to use ES instead of the absolute or relative difference between predicted and actual 1RM? Please explain

Comment 7: In lines 151 to 154, the criterion for Pearson's r correlation coefficients was given, but correlation is not mentioned in the results or in the discussion, so it seems redundant. What is the role of Pearson's r, please explain.

Comment 8: In line 155 you state that discrepancies between reviewers were resolved through discussion. It is not clear what discrepancies were resolved, please describe.

Comment 9: What statistical methods were used to calculate missing values (line 160)? please describe in more detail.

Comment 10: Please describe the rationale for using the Downs and Black checklist for quality assessment of reliability/validity studies and its comparison with the PEDro scale.

Results

Comment 11: Because test-retest reliability might decrease with time between measurements (i.e., the ICC within session is usually greater than between sessions), it would be informative to report the time between two consecutive measurements in the included studies.

Discussion

Comment 12: The authors refer to Figure 2 and conclude that the two-point load-velocity method is a valid approach for predicting 1RM based on ES. Looking at Figure 2, it can be seen that some studies underestimated the 1RM, resulting in negative ES. How were these accounted for in the analyses? The methods of the overall ES are not well described, so it is not clear what the final value actually represents. A large underestimate could offset an overestimate, resulting in no bias, although the bias is actually present. The authors themselves outline this problem in lines 401 to 413, where IMU overestimates and underestimates the 1RM in two studies, but the synthesized ES gives -0.09 [95%CI: -0.670, 0.489]. Please explain the methodology in detail and reconsider the use of non-negative ES or relative bias (%). In case of different results please change the discussion accordingly.

Comment 13. Line 318; reconsider the methodology for quantifying the validity (dealing with negative ES) of studies based on the exercise selection factor.

Reviewer #2: General comment

Authors conducted an interesting review which is generally well-written. I have provided several specific comments that should be considered before the formal acceptance of the manuscript.

Specific comments

Line 24. Change to “p < 0.001”.

Line 25. “of two-point method” can be deleted.

Line 26. Instead of indicating “a few” it is better that authors report the exact number of studies.

Line 27. “the reliability of it” should be replaced by “its reliability”.

Line 35. At the end of the first sentence before “Moreover”, the “,” should be replaced by “.”.

Line 84. Great introduction!

Line 117. Inclusions criteria 6 and 6 can be grouped together.

Line 124. Use the abbreviation LVR consistently through the manuscript.

Line 144. This is only of the statistics that can be used but it is problematic. Imagine subject A (1RM = 100 kg for direct method and 150 kg for 2-point method) and subject B (1RM = 150 kg for direct method and 100 kg for 2-point method). The effect size would be 0.00 suggesting an excellent validity, but the error was of 50 kg for both subjects (not accurate procedure). This should be highlighted as a limitation of the study. Because of these problems most of the studies report the absolute errors between 1RM testing procedures.

Line 172. “2014” should be deleted.

Table 1. Year of publication should be deleted in the table (same applies for different tables).

Line 190. Did you contact corresponding author? Anyway, the citation of this study should be included in this sentence to make aware of the readers about this paper.

Line 195. Write full words for M and SD. These abbreviations were not introduced before.

Line 231. Different types of regression models usually refer to linear pr polynomial models but this is not the case for the 2-point method because only the linear model can be used. Authors should use other term for this.

Line 261. The moderate level of reliability is likely caused because it has been considered two-point methods in which the loads were very close or very far from the 1RM. Besides this overall ICC value, authors are suggested to report also the ICC value obtained in each study considering only the 2-point method constructed considering the load closer to the 1RM in order to provide practitioners a more precise indicator of the ICC when the 2-point method is conducted as it is supposed to be conducted.

Line 283. Why both types of effect sizes were calculated? Is this a typo?

Line 286. Use the abbreviation “ES” consistently through the manuscript.

Line 309. The reason is VERY CLEAR. The 2-point methods with low ICC are because the 2 loads are very close (20%1RM difference). Please, refer to the following article to see why this is problematic (https://journals.humankinetics.com/view/journals/ijspp/aop/article-10.1123-ijspp.2023-0127/article-10.1123-ijspp.2023-0127.xml; see specifically the section “Distance Between the 2 Extreme Experimental Points”). Other 2-point methods are more reliable because the 2 loads are more distant. This should be highlighted in the manuscript and, as I said before, I think it is important to also consider from each study only the 2-point method that was applied following the guidelines recently highlighted by García-Ramos (https://journals.humankinetics.com/view/journals/ijspp/aop/article-10.1123-ijspp.2023-0127/article-10.1123-ijspp.2023-0127.xml). It is OK if you report the overall ICC, but a new ICC should be provided to highlight which is the expected ICC when the 2-point method is applied as it is supposed to be applied.

Line 311. It is obvious that the 2-point method is reliable based on your data, but you should not use for this conclusion 2-point methods that select loads that are not appropriated and that no one would/should use in practice.

Line 339. Change “plane” by “direction”.

Line 341. Use the abbreviation LVR consistently!

Line 345. None of these studies are assessing neuromuscular adaptations, they are assessing EMG in an acute study. Longitudinal studies do not support this sentence:

- https://pubmed.ncbi.nlm.nih.gov/37535335/

- https://pubmed.ncbi.nlm.nih.gov/37340878/

Line 357. Can you present the comparative of ES that supports this sentence? Maybe it is because the SSC has been mainly used with lower-body exercise and the CO with upper-body-exercise. If this is the case, how do we know that the mentioned differences between exercise technique are not caused by the exercise? I do not think the SSC is problematic when using the 2-point method, I think authors are confused by the mix of exercises. The absolute errors of individual studies directly comparing concentric-only and eccentric-concentric should be analysed to make this type of statement in the paper.

Line 375. These studies used generalized load-velocity relationship which are different to the individualized load-velocity relationship analysed in this study. Please, see this paper for the problems of generalized load-velocity relationship (https://www.thieme-connect.com/products/ejournals/abstract/10.1055/a-2158-3848). Not sure if providing information about general LVR is important here.

Line 381. There are MANY more exceptions as highlighted in the paper mentioned in the previous comment.

Line 423. Please, consider this paper here: https://pubmed.ncbi.nlm.nih.gov/33475985/

Line 461. Agree! Some of these problems have been addressed here recently: https://journals.humankinetics.com/view/journals/ijspp/aop/article-10.1123-ijspp.2023-0127/article-10.1123-ijspp.2023-0127.xml

Line 518. RT has been abbreviated multiple times in the manuscript. Please, carefully check the paper to avoid these errors.

Line 531. Why should we expect different results with women? As far as they have RT experience as men, they should be no problem at all.

Line 535. It does not require further investigation, the problem is considering 2-point methods based on two loads that are far from the optimal ones.

Line 541. Not sure it is inconclusive, clear guidelines have been provided here: https://journals.humankinetics.com/view/journals/ijspp/aop/article-10.1123-ijspp.2023-0127/article-10.1123-ijspp.2023-0127.xml

Line 545. I agree, but the problem of the systematic review is that we analysing the effect of one factor, there are differences in more factors (e.g., execution technique when one technique is more used with lower-body exercises and other technique more used with upper-body exercise). Therefore, these conclusions should be based in studies that only manipulate 1 condition, whereas the rest of the factors remain fixed. This should be specified as another limitation of the systematic review.

Line 547. Practical applications are OK, but authors are repeating several times the same information. Please, synthesize the paragraph.

6. PLOS authors have the option to publish the peer review history of their article (what does this mean?). If published, this will include your full peer review and any attached files.

Reviewer #1: No

Reviewer #2: No

---

## [Author Response · Author response to Decision Letter 0]

22 Sep 2023

Dear Editors and Reviewers:

 Thank you for your comments concerning our manuscript entitled “Is two-point method a valid and reliable method to predict 1RM? A systematic review”. We gratefully thank you for your comments and professional advice, which has significantly raised the quality of the manuscript and has enable us to improve the manuscript. We have studied comments carefully and have made correction which we hope meet with approval.

Responds to the reviewer’s comments: 

Reviewer #1:

General comment: 

First, I would like to congratulate the authors for their efforts on the manuscript “Is two-point method a valid and reliable method to predict 1RM? A systematic review”. The article addresses the prediction of the maximum load that can be lifted once with a proficient technique (1RM) using the load-velocity relationship.

The topic is particularly important because in recent years, velocity-based training has gained popularity among strength and conditioning coaches. However, methodological considerations for implementing the two-point load-velocity method to assess 1RM are scarce. In the first part of this systematic review, the reliability and validity of 1RM predicted with load-velocity relationship are analysed. In the second part, the authors provide practical advice on how to perform the two-point method to obtain the most reliable and valid 1RM by explaining the factors that most affect the accuracy of the method. The authors conclude that the two-point load-velocity method is reliable and valid for obtaining the 1RM, outlining the selection of the appropriate velocity and regression models as the most important factors.

One of the most important considerations of this review is the statistical method used to assess the overall validity of the two-point method from the included studies. For each study separately, the authors calculated the difference between the predicted and direct 1RM as the effect size (ES). Based on the average value from all studies, the synthesized validity of the method was assessed. However, using the average values from the studies that had overestimates and under-estimations could lead to an erroneous bias. I suggest this should be corrected as the result could change the interpretation and conclusions. Detailed comments with minor considerations and suggestions can be found as comments in the text below. 

Reply (General comment): Thank you for your professional comments on our manuscript, we have carefully considered each one. Detailed responses were presented below.

Comment 1: Lines 92 to 95 indicate the search strategy that led to 21, 28, 29, and 9 sources at the identification level. This is a relatively small number of studies for the first level of search and the main consideration is that important literature was omitted due to the very specific terms used. Please describe why you chose to use such a search strategy.

Reply (Comment 1): Thank you for your valuable advice on this issue. In 2017, the two-point method was first introduced to LVR to simplify the LVR testing process. Subsequently, the two-point method was also gradually used to determine 1RM. Therefore, two-point LVR is a new research field, and there is not much research literature in this field at present. After a careful review of several key articles, we found that researchers were happy to emphasize the use of the "two-point method" in the title or abstract to highlight the focus of their research. Therefore, in order to conduct a comprehensive search for articles with 1RM determination using two-point LVR, we set the following keywords for literature screening: ("two-point method*" OR "two-point" OR "2-point method*" OR "2-point" OR "two-load method*" OR "two loads" OR "2-load method*" OR "2 loads") AND ("one-repetition maximum" OR "1RM" OR "maximal dynamic strength" OR "1-RM"). At the beginning of the project, our research group considered "LVR" and "1RM" as the two key words for literature search, because LVR is a more macroscopic concept than the two-point method. However, the use of the two keywords "two-point method" and "1 RM" was considered more appropriate for the subjects in this study than "LVR" and "1 RM", and the use of "LVR" and "1RM" did not add more included articles. Therefore, under comprehensive consideration, we chose "two-point method" and "1RM" as the keywords of literature search.

Comment 2: Under line 113 the fifth (5)) Selection criteria were studies that used the two-point method to predict 1RM, with the exception of L0 and F0. I recommend that the criteria used to exclude the data (lines 114 to 116) be transferred to the Data Extraction section.

Reply (Comment 2): Thank you for your suggestion, we have made adjustments in the resubmitted manuscript.

Comment 3: When describing F0 (load at zero velocity) in line 114, it is better to use force at zero velocity.

Reply (Comment 3): We feel sorry for our carelessness. In our resubmitted manuscript, the typo is revised. Thanks for your correction.

Comment 4: The ES of the differences between the predicted and actual 1RM were calculated. It is not clear from the text (lines 135 to 138) exactly how this was done. Please provide the equation for calculating the ES and include the references for this statistical procedure.

Reply (Comment 4): Thank you for pointing out the problem in the manuscript. Because we did not make it clear, you have misunderstood the content of the manuscript. If the included literature reported the ES of the difference between the predicted 1RM and the actual 1RM, the ES value reported in the literature was extracted directly, if not, it was calculated by the formula “the mean difference divided by the between-subject SD”, which was derived from the literature “DOI: 10.1249/MSS.0b013e31818cb278”. At the same time, we have changed the language in the manuscript so that readers do not misunderstand the meaning of the manuscript.

Comment 5: You mention the total ES later in the text of the manuscript. However, it is not clear how this total effect was calculated using the methods. Please describe this.

Reply (Comment 5): Thank you for your question. On lines 146-147 of the manuscript, we describe the synthesis of the total ES, but it may not be placed in the right place and it is easy to ignore this part of the text. Therefore, in the new manuscript, we put the synthetic method of total ES in a more appropriate position. The total ES was calculated by a software named “Comprehensive Meta-Analysis V3 software (Biostat, New Jersey, USA)”. In the software, a total ES value is automatically generated by entering the mean, standard deviation, and sample size of the predicted 1RM and actual 1RM for each included study. This software is also commonly used in other systematic reviews or meta-analyses of synthetic effect sizes.

Comment 6: Why did you choose to use ES instead of the absolute or relative difference between predicted and actual 1RM? Please explain.

Reply (Comment 6): Thank you for your comments. We chose to use ES to reflect the predicted 1RM and the actual 1RM because ES is a good indicator of the degree of difference between the two, and because ES can be compared between different studies, it is easier to synthesize ES when summarizing all the literature, as mentioned in response 5. However, when absolute differences are used, it may be difficult to compare between studies due to the large variation in 1RM between different movements, e.g. when the absolute difference between the predicted 1RM and the actual 1RM is 10kg, the accuracy of upper limb exercises is lower than that of lower limb exercises because lower limb strength is usually greater. The relative difference between the predicted 1RM and the actual 1RM seems to be a good indicator, but since some of the included literatures did not report this value, it needs to be indirectly calculated by the formula. Since the data distribution of the included articles is not known, the estimation of relative difference will cause more error than the calculation of ES. Based on the team's overall consideration, we chose ES to reflect the degree of difference between the predicted 1RM and the actual 1RM.

Comment 7: In lines 151 to 154, the criterion for Pearson's r correlation coefficients was given, but correlation is not mentioned in the results or in the discussion, so it seems redundant. What is the role of Pearson's r, please explain.

Reply (Comment 7): Thank you for pointing out the problem in the manuscript. The Pearson correlation coefficient (r) was extracted from the included literature to reflect the degree of correlation between the predicted 1RM and the actual 1RM, i.e., the higher the accuracy of the two-point method in predicting 1RM, the closer r approaches 1. We have extracted the r values reported in the included studies into Table 3, but unfortunately, due to our oversight, the interpretation of r is not detailed in the discussion section. In our latest manuscript, we have explained the interpretation of the r value in detail in the discussion section. Thank you again for pointing out the shortcomings of our manuscript, which was very helpful in improving the quality of this study.

Comment 8: In line 155 you state that discrepancies between reviewers were resolved through discussion. It is not clear what discrepancies were resolved, please describe.

Reply (Comment 8): Thank you for your question. At the beginning of this study, there was some controversy among the members of the research team about which indicators to extract, such as whether to select the effect size as a validity indicator to evaluate the accuracy of the two-point method in predicting 1RM or the relative difference. After the in-depth discussion of the research group, we finally reached a consensus. We have added relevant explanations in the latest submitted manuscript.

Comment 9: What statistical methods were used to calculate missing values (line 160)? please describe in more detail.

Reply (Comment 9): Thank you for your question. For data where actual 1RM was provided and absolute difference was provided, we calculated missing values (predicted 1RM) using “M_(1+2)=M_1+M_2, 〖SD〗_(1+2)=√(〖SD〗_1^2+〖SD〗_2^2 )”. For data where 1RM was provided for both males and females, but not for the overall 1RM, we used “M_(1&2)=(∑▒〖n_i X ®_i 〗)/(∑▒n_i ), 〖SD〗_(1&2)=√((∑▒〖n_i s_i^2+∑▒〖n_i d_i^2 〗〗)/(∑▒n_i ))” to calculate missing values. In the meantime, we have made a detailed supplementary explanation in the resubmitted manuscript.

Comment 10: Please describe the rationale for using the Downs and Black checklist for quality assessment of reliability/validity studies and its comparison with the PEDro scale.

Reply (Comment 10): Thank you for your question. The modified version of the Downs and Black checklist was used to assess the quality of the included articles, which consisted of ten questions:

Number Question

Q1 Is the hypothesis/aim/objective of the study clearly described?

Q2 Are the main outcomes to be measured clearly described in the Introduction or Methods section? If the main outcomes are first mentioned in the Results section, the question should be answered no.

Q3 Are the characteristics of the patients included in the study clearly described? In cohort studies and trials, inclusion and/or exclusion criteria should be given. In case‐control studies, a case‐definition and the source for controls should be given.

Q6 Are the main findings of the study clearly described? Simple outcome data (including denominators and numerators) should be reported for all major findings so that the reader can check the major analyses and conclusions. (This question does not cover statistical tests which are considered below).

Q7 Does the study provide estimates of the random variability in the data for the main outcomes? 

In non normally distributed data the inter‐quartile range of results should be reported. In normally distributed data the standard error, standard deviation or confidence intervals should be reported. If the distribution of the data is not described, it must be assumed that the estimates used were appropriate and the question should be answered yes.

Q10 Have actual probability values been reported (e.g. 0.035 rather than <0.05) for the main outcomes except where the probability value is less than 0.001?

Q11 Were the subjects asked to participate in the study representative of the entire population from which they were recruited? The study must identify the source population for patients and describe how the patients were selected. Patients would be representative if they comprised the entire source population, an unselected sample of consecutive patients, or a random sample. Random sampling is only feasible where a list of all members of the relevant population exists. Where a study does not report the proportion of the source population from which the patients are derived, the question should be answered as unable to determine.

Q16 If any of the results of the study were based on “data dredging”, was this made clear? Any analyses that had not been planned at the outset of the study should be clearly indicated. If no retrospective unplanned subgroup analyses were reported, then answer yes.

Q18 Were the statistical tests used to assess the main outcomes appropriate? The statistical techniques used must be appropriate to the data. For example nonparametric methods should be used for small sample sizes. Where little statistical analysis has been undertaken but where there is no evidence of bias, the question should be answered yes. If the distribution of the data (normal or not) is not described it must be assumed that the estimates used were appropriate and the question should be answered yes.

Q20 Were the main outcome measures used accurate (valid and reliable)? For studies where the outcome measures are clearly described, the question should be answered yes. For studies which refer to other work or that demonstrates the outcome measures are accurate, the question should be answered as yes.

 In fact, the original version of the Downs and Black checklist contained a total of 27 questions (doi:10.1136/jech.52.6.377). We did not consider using the original version because it contains many question options for intervention, and involved studies did not involve exercise intervention. We selected ten questions closely related to this review to evaluate the quality of the included studies. However, this modified version of the Downs and Black checklist did not have a suitable classification of literature quality. We found that the PEDro scale also has a total score of 10, with high quality being 9-10, medium quality 5-8, and low quality 0-4. In terms of scores, the modified version of the Downs and Black checklist used in this review was well suited for analogy with the PEDro scale. Therefore, we considered the study with a score of 9-10 to be of high quality, 5-8 to be of medium quality, and 0-4 to be of low quality.

Comment 11: Because test-retest reliability might decrease with time between measurements (i.e., the ICC within session is usually greater than between sessions), it would be informative to report the time between two consecutive measurements in the included studies.

Reply (Comment 11): Thank you for pointing this out in our article. We have extracted the retest intervals from the included literature in the resubmitted manuscript.

Comment 12: The authors refer to Figure 2 and conclude that the two-point load-velocity method is a valid approach for predicting 1RM based on ES. Looking at Figure 2, it can be seen that some studies underestimated the 1RM, resulting in negative ES. How were these accounted for in the analyses? The methods of the overall ES are not well described, so it is not clear what the final value actually represents. A large underestimate could offset an overestimate, resulting in no bias, although the bias is actually present. The authors themselves outline this problem in lines 401 to 413, where IMU overestimates and underestimates the 1RM in two studies, but the synthesized ES gives -0.09 [95%CI: -0.670, 0.489]. Please explain the methodology in detail and reconsider the use of non-negative ES or relative bias (%). In case of different results please change the discussion accordingly.

Reply (Comment 12): Thank you for pointing out this issue in the manuscript. ES represents the difference between the predicted 1RM and the actual 1RM. The closer the ES is to 0, the better the prediction of 1RM by the two-point method. The overall ES was calculated by the "Comprehensive Meta-Analysis V3 software (Biostat, New Jersey, USA)" software. We are sorry to note that we have used an inappropriate statistical method to overestimate or underestimate 1RM as a measure of the accuracy of the two-point method in predicting 1RM, which has led to bias in the summary of ES. We have adjusted all ES values to non-negative ES values, recalculated the total ES, and re-discussed the new results in the revised submission.

Comment 13: Line 318; reconsider the methodology for quantifying the validity (dealing with negative ES) of studies based on the exercise selection factor.

Reply (Comment 13): Thank you for your question. We also converted negative ES values to non-negative ES values and recalculated the total ES.

Reviewer #2:

General comment: Authors conducted an interesting review which is generally well-written. I have provided several specific comments that should be considered before the formal acceptance of the manuscript.

Reply (General comment): Thank you for your approval of our manuscript. We have revised the manuscript in detail according to your comments.

Comment 1: Line 24. Change to “p < 0.001”.

Reply (Comment 1): Thank you for your suggestions, we have made corresponding changes in the manuscript.

Comment 2: Line 25. “of two-point method” can be deleted.

Reply (Comment 2): Thank you for your suggestions. We have deleted this from the revised manuscript.

Comment 3: Line 26. Instead of indicating “a few” it is better that authors report the exact number of studies.

Reply (Comment 3): Thank you for your suggestions. We have reported the specific number of studies in the revised manuscript.

Comment 4: Line 27. “the reliability of it” should be replaced by “its reliability”.

Reply (Comment 4): Thank you for your suggestions. We have made corresponding changes in the manuscript.

Comment 5: Line 35. At the end of the first sentence before “Moreover”, the “,” should be replaced by “.”.

Reply (Comment 5): We were really sorry for our careless mistakes. Thank you for your reminder.

Comment 6: Line 84. Great introduction!

Reply (Comment 6): Thank you for your approval.

Comment 7: Line 117. Inclusions criteria 6 and 6 can be grouped together.

Reply (Comment 7): Thank you for your suggestion, we have made corresponding adjustments in the revised manuscript.

Comment 8: Line 124. Use the abbreviation LVR consistently through the manuscript.

Reply (Comment 8): We were really sorry for our careless mistakes. We have made corresponding changes in the latest submitted manuscript.

Comment 9: Line 144. This is only of the statistics that can be used but it is problematic. Imagine subject A (1RM = 100 kg for direct method and 150 kg for 2-point method) and subject B (1RM = 150 kg for direct method and 100 kg for 2-point method). The effect size would be 0.00 suggesting an excellent validity, but the error was of 50 kg for both subjects (not accurate procedure). This should be highlighted as a limitation of the study. Because of these problems most of the studies report the absolute errors between 1RM testing procedures.

Reply (Comment 9): Thank you for pointing out this deficiency of the study, which may seriously affect the quality of this review. We have converted all ES to non-negative ES in the latest submitted manuscript to avoid bias in the results.

Comment 10: Line 172. “2014” should be deleted.

Reply (Comment 10): Thank you for your suggestions. We have deleted this from the revised manuscript.

Comment 11: Table 1. Year of publication should be deleted in the table (same applies for different tables).

Reply (Comment 11): Thank you for your suggestions. We have made corresponding changes in the manuscript.

Comment 12: Line 190. Did you contact corresponding author? Anyway, the citation of this study should be included in this sentence to make aware of the readers about this paper.

Reply (Comment 12): At the time of the initial submission, we had a potentially eligible study that was not available, even though we had sent an application email to the corresponding author. Recently, however, we obtained the full text of this article, and after our review, we found that this article did meet the inclusion criteria for this review, so we included this study in the latest submission.

Comment 13: Line 195. Write full words for M and SD. These abbreviations were not introduced before.

Reply (Comment 13): Thank you for raising this issue in the manuscript. We have made corresponding changes in the revised submission.

Comment 14: Line 231. Different types of regression models usually refer to linear pr polynomial models but this is not the case for the 2-point method because only the linear model can be used. Authors should use other term for this.

Reply (Comment 14): Thank you for your comment. We have changed "Different types of regression models" to "different methods of 1RM determination based on two-point method" in the revised manuscript.

Comment 15: Line 261. The moderate level of reliability is likely caused because it has been considered two-point methods in which the loads were very close or very far from the 1RM. Besides this overall ICC value, authors are suggested to report also the ICC value obtained in each study considering only the 2-point method constructed considering the load closer to the 1RM in order to provide practitioners a more precise indicator of the ICC when the 2-point method is conducted as it is supposed to be conducted.

Reply (Comment 15): Thank you for your comment. We have given your suggestion serious consideration. We realized that the poor choice of test loads (the small difference between the two test loads) was driving down the overall ICC and we really needed to report this. However, considering the structure of the chapter, we feel that it seems more appropriate to discuss this issue in detail in the "discussion section", so we do not continue to add it in the "results section". Thanks again.

Comment 16: Line 283. Why both types of effect sizes were calculated? Is this a typo?

Reply (Comment 16): Thank you for your comment. Because some of the included articles reported Cohen's d and some reported Hedge's g, we meant to use "OR" to link the two words, but because of our mistake, we used "AND" to mislead you. After much deliberation, we decided to delete this note from the revised manuscript so as not to unnecessarily mislead the reader.

Comment 17: Line 286. Use the abbreviation “ES” consistently through the manuscript.

Reply (Comment 17): Thank you for your comment. We have changed the effect size except for the first occurrence to ES in the revised manuscript submission.

Comment 18: Line 309. The reason is VERY CLEAR. The 2-point methods with low ICC are because the 2 loads are very close (20%1RM difference). Please, refer to the following article to see why this is problematic (https://journals.humankinetics.com/view/journals/ijspp/aop/article-10.1123-ijspp.2023-0127/article-10.1123-ijspp.2023-0127.xml; see specifically the section “Distance Between the 2 Extreme Experimental Points”). Other 2-point methods are more reliable because the 2 loads are more distant. This should be highlighted in the manuscript and, as I said before, I think it is important to also consider from each study only the 2-point method that was applied following the guidelines recently highlighted by García-Ramos (https://journals.humankinetics.com/view/journals/ijspp/aop/article-10.1123-ijspp.2023-0127/article-10.1123-ijspp.2023-0127.xml). It is OK if you report the overall ICC, but a new ICC should be provided to highlight which is the expected ICC when the 2-point method is applied as it is supposed to be applied.

Reply (Comment 18): Thank you for your comments. We have carefully read the literature you have recommended and have incorporated the relevant arguments in this section into the revised manuscript. After removing the unreasonable load selection, we found that the overall ICC was equal to 0.911 (excellent), which was a very reliable level. You mentioned in your comment that only two loads should be tested, but we found that this was not done in the included studies, so we wrote this point into the "future research direction in two-point method".

Comment 19: Line 311. It is obvious that the 2-point method is reliable based on your data, but you should not use for this conclusion 2-point methods that select loads that are not appropriated and that no one would/should use in practice.

Reply (Comment 19): Thank you for your comments. You pointed out the inadequacies of our manuscript in your previous comments, and we have made the corresponding changes and deleted this unreasonable content.

Comment 20: Line 339. Change “plane” by “direction”.

Reply (Comment 20): Thank you for raising this issue in the manuscript. We have made corresponding changes in the revised submission.

Comment 21: Line 341. Use the abbreviation LVR consistently!

Reply (Comment 21): Thank you for raising this issue in the manuscript. We have made corresponding changes in the revised submission.

Comment 22: Line 345. None of these studies are assessing neuromuscular adaptations, they are assessing EMG in an acute study. Longitudinal studies do not support this sentence:

- https://pubmed.ncbi.nlm.nih.gov/37535335/

- https://pubmed.ncbi.nlm.nih.gov/37340878/

Reply (Comment 22): Thank you for your comments. After carefully reading the two articles you recommended, we realized that what we described in the original submitted manuscript was not appropriate. However, from practical experience, athletes prefer to use free-weight for RT rather than Smith machine.

Comment 23: Line 357. Can you present the comparative of ES that supports this sentence? Maybe it is because the SSC has been mainly used with lower-body exercise and the CO with upper-body-exercise. If this is the case, how do we know that the mentioned differences between exercise technique are not caused by the exercise? I do not think the SSC is problematic when using the 2-point method, I think authors are confused by the mix of exercises. The absolute errors of individual studies directly comparing concentric-only and eccentric-concentric should be analysed to make this type of statement in the paper.

Reply (Comment 23): Thank you for your professional comment. In our analysis of this, we have synthesized all the exercises without taking into account the problem you have mentioned. We have carefully considered your suggestion and have only analyzed CO and EC for the same exercise in the revised submission. Thank you again.

Comment 24: Line 375. These studies used generalized load-velocity relationship which are different to the individualized load-velocity relationship analysed in this study. Please, see this paper for the problems of generalized load-velocity relationship (https://www.thieme-connect.com/products/ejournals/abstract/10.1055/a-2158-3848). Not sure if providing information about general LVR is important here.

Reply (Comment 24): Thank you for your comment. We did not find a suitable argument (individual LVR or two-point LVR) to discuss this issue, so we cited the literature on general LVR and mentioned possible future research directions in the following description of the manuscript.

Comment 25: Line 381. There are MANY more exceptions as highlighted in the paper mentioned in the previous comment.

Reply (Comment 25): Thank you for your comment. We have cited more exceptions from your recommended article in the revised submission.

Comment 26: Line 423. Please, consider this paper here: https://pubmed.ncbi.nlm.nih.gov/33475985/

Reply (Comment 26): Thank you for your advice. We have carefully read the literature you recommended and have made corresponding modifications and citations in the revised manuscript.

Comment 27: Line 461. Agree! Some of these problems have been addressed here recently: https://journals.humankinetics.com/view/journals/ijspp/aop/article-10.1123-ijspp.2023-0127/article-10.1123-ijspp.2023-0127.xml

Reply (Comment 27): Thank you for your comments. After a careful reading of the literature you recommended, we found that some studies have applied the two-point method to field conditions, but it seems that no studies have involved the prediction of 1RM. We believe that there will be more studies to investigate whether the two-point method can accurately predict 1RM of RT in field conditions.

Comment 28: Line 518. RT has been abbreviated multiple times in the manuscript. Please, carefully check the paper to avoid these errors.

Reply (Comment 28): We have carefully checked the manuscript and corrected the errors accordingly.

Comment 29: Line 531. Why should we expect different results with women? As far as they have RT experience as men, they should be no problem at all.

Reply (Comment 29): Thank you for your comments. We did not consider it sufficiently, and we have deleted it from the manuscript at your suggestion. At the same time, since we extracted the prediction of 1RM by two-point method under fatigue and non-fatigue conditions from the newly included literature, we listed "fatigue state" as a new potential influencing factor.

Comment 30: Line 535. It does not require further investigation, the problem is considering 2-point methods based on two loads that are far from the optimal ones.

Reply (Comment 30): Thank you for your comments. Indeed, as you said, the reason for the low ICC was due to the unreasonable load selection (the difference between the two test loads was too small), so we changed this description in the revised manuscript.

Comment 31: Line 541. Not sure it is inconclusive, clear guidelines have been provided here: https://journals.humankinetics.com/view/journals/ijspp/aop/article-10.1123-ijspp.2023-0127/article-10.1123-ijspp.2023-0127.xml

Reply (Comment 31): Thank you for your comment. We have revised the manuscript based on the references you suggested.

Comment 32: Line 545. I agree, but the problem of the systematic review is that we analysing the effect of one factor, there are differences in more factors (e.g., execution technique when one technique is more used with lower-body exercises and other technique more used with upper-body exercise). Therefore, these conclusions should be based in studies that only manipulate 1 condition, whereas the rest of the factors remain fixed. This should be specified as another limitation of the systematic review.

Reply (Comment 32): Thank you for your comment. We have described the limitations of this study in a new manuscript with your comments.

Comment 33: Line 547. Practical applications are OK, but authors are repeating several times the same information. Please, synthesize the paragraph.

Reply (Comment 33): Thank you for your comments. We have simplified the Practical applications section appropriately.

---

## [Decision Letter · Decision Letter 1]

3 Nov 2023

Is two-point method a valid and reliable method to predict 1RM? A systematic review

PONE-D-23-21953R1

Dear Dr. Xiuli Zhang,

We’re pleased to inform you that your manuscript has been judged scientifically suitable for publication and will be formally accepted for publication once it meets all outstanding technical requirements. Just please make a final minor modification of the abstract as sugested by the reviewer 2 (see comment below). 

Kind regards,

Danica Janicijevic, Ph.D

Academic Editor

PLOS ONE

Additional Editor Comments (optional):

Reviewers' comments:

Reviewer's Responses to Questions

**Comments to the Author**

1. If the authors have adequately addressed your comments raised in a previous round of review and you feel that this manuscript is now acceptable for publication, you may indicate that here to bypass the “Comments to the Author” section, enter your conflict of interest statement in the “Confidential to Editor” section, and submit your "Accept" recommendation.

Reviewer #1: All comments have been addressed

Reviewer #2: All comments have been addressed

2. Is the manuscript technically sound, and do the data support the conclusions?

Reviewer #1: Yes

Reviewer #2: Yes

3. Has the statistical analysis been performed appropriately and rigorously? 

Reviewer #1: Yes

Reviewer #2: Yes

4. Have the authors made all data underlying the findings in their manuscript fully available?

Reviewer #1: Yes

Reviewer #2: Yes

5. Is the manuscript presented in an intelligible fashion and written in standard English?

Reviewer #1: Yes

Reviewer #2: Yes

6. Review Comments to the Author

Reviewer #1: (No Response)

Reviewer #2: I appreciated the detailed responses provide by the authors and I believe the manuscript is much stronger after consideing the comments from both reviewers. I believe the paper is ready for publication. Only one aspect that should be modified. In the abstract authors indicated:

"The findings of this review indicated that the two-point method slightly overestimated 1RM (effect size = 0.203 [95%CI: 0.132, 0.275]; P < 0.001)". However, in this new version of the manuscript authors also indicated "We used non-negative ES to reflect the degree of difference between the predicted 1RM and the actual 1RM to avoid bias in the results". Therefore, authors cannot say that the 1RM was OVERESTIMATED, they can only indicate that small differences (ES = 0.20) exist but it is not possible to indicate here the direction of the differences because negative effect size (lower 1RM estimated by 2-point method) were converted to positive effect size.

After this minor change is made, I believe the paper can be accepted for publication.

7. PLOS authors have the option to publish the peer review history of their article (what does this mean?). If published, this will include your full peer review and any attached files.

Reviewer #1: No

Reviewer #2: **Yes: **Amador García Ramos

---

## [Editor Report · Acceptance letter]

7 Nov 2023

PONE-D-23-21953R1 

Is two-point method a valid and reliable method to predict 1RM? A systematic review 

Dear Dr. Zhang:

I'm pleased to inform you that your manuscript has been deemed suitable for publication in PLOS ONE. Congratulations! Your manuscript is now with our production department. 

Kind regards, 

on behalf of

Dr. Danica Janicijevic 

Academic Editor

PLOS ONE